# Hypoxia induces HIF1α-dependent epigenetic vulnerability in triple negative breast cancer to confer immune effector dysfunction and resistance to anti-PD-1 immunotherapy

Shijun Ma [1], Yue Zhao[2], Wee Chyan Lee[1], Li-Teng Ong[1], Puay Leng Lee[1], Zemin Jiang[1], Gokce Oguz[1], Zhitong Niu[3], Min Liu[2], Jian Yuan Goh[1], Wenyu Wang [3], Matias A. Bustos[4], Sidse Ehmsen [5], Adaikalavan Ramasamy[1], Dave S. B. Hoon[4], Henrik J. Ditzel [5,6], Ern Yu Tan[7], Qingfeng Chen [2✉] & Qiang Yu [1,8,9✉]

The hypoxic tumor microenvironment has been implicated in immune escape, but the underlying mechanism remains elusive. Using an in vitro culture system modeling human T cell dysfunction and exhaustion in triple-negative breast cancer (TNBC), we find that hypoxia suppresses immune effector gene expression, including in T and NK cells, resulting in immune effector cell dysfunction and resistance to immunotherapy. We demonstrate that hypoxia-induced factor 1α (HIF1α) interaction with HDAC1 and concurrent PRC2 dependency causes chromatin remolding resulting in epigenetic suppression of effector genes and subsequent immune dysfunction. Targeting HIF1α and the associated epigenetic machinery can reverse the immune effector dysfunction and overcome resistance to PD-1 blockade, as demonstrated both in vitro and in vivo using syngeneic and humanized mice models. These findings identify a HIF1α-mediated epigenetic mechanism in immune dysfunction and provide a potential strategy to overcome immune resistance in TNBC.

[1] Genome Institute of Singapore, Agency for Science, Technology and Research (A*STAR), Singapore 138672, Singapore. [2] Institute of Molecular and Cellular Biology, Agency for Science, Technology and Research (A*STAR), Singapore 138673, Singapore. [3] The Sixth Affiliated Hospital of Sun Yat-sen University, Guangzhou 510655, China. [4] Department of Translational Molecular Medicine, Saint John's Cancer Institute, Providence Health System, Santa Monica, CA 90404, USA. [5] Department of Oncology, Odense University Hospital, Odense 5230, Denmark. [6] Department of Cancer and Inflammation Research, Institute of Molecular Medicine, University of Southern Denmark, Odense 5230, Denmark. [7] Department of General Surgery, Tan Tock Seng Hospital, Singapore 308433, Singapore. [8] Department of Physiology, Yong Loo Lin School of Medicine, National University of Singapore, Singapore 117597, Singapore. [9] Cancer and Stem Cell Biology, Duke-NUS Medical School, Singapore 169857, Singapore. ✉email: qchen@imcb.a-star.edu.sg; yuq@gis.a-star.edu.sg

Triple-negative breast cancer (TNBC) is an aggressive subtype of breast cancer that lacks effective treatment[1,2]. Paradoxically, despite strong immunogenicity and a higher prevalence of tumor-infiltrating lymphocytes (TILs) in TNBC[3], patients often progress to metastasis with poor clinical outcomes[1]. This indicates a dysfunctional tumor immune microenvironment in TNBC. Immune checkpoint blockade (ICB) directed against programmed death-1 (PD-1) or programmed death-ligand 1 (PD-L1) has shown promising results in TNBC[4–6], but the overall response rate of ICB is less than 20%[7]. The tumor microenvironment (TME) consists of diverse components that provide multiple cellular and molecular factors to impact the effector functions of TILs, leading to immune evasion and resistance to immunotherapy. However, the fundamental mechanism governing tumor immune resistance remains unanswered.

Hypoxia is a hallmark of most solid tumors and a common trait of TME that is associated with cancer progression and metastasis[8–10]. In breast cancer, hypoxia is more evident in TNBC than in other breast cancer subtypes[11,12]. Considerable evidence in mouse models has shown that hypoxia promotes tumor escape from immune surveillance and immunotherapy[13–19]. In addition, there is also increasing clinical evidence suggesting a role of tumor hypoxia in the exclusion of immune cell tumor infiltration and resistance to ICB therapy[20,21], yet the mechanisms remain to be determined. Nevertheless, there are also contradictory reports showing the role of hypoxia in potentiating the activity of cytotoxic T cells, though these studies were mainly conducted in mouse models[22–24]. Therefore, how hypoxia affects human immune cells in modulating antitumor immunity remains clarified.

Tumor-infiltrating T cell terminal exhaustion resulting from chronic and continuous tumor antigen stimulation is the primary factor contributing to immune evasion and resistance to cancer immunotherapy[25]. In the present study, we sought to use an in vitro system to model the effect of hypoxia on the immune cell activity in TME and investigate the underlying mechanism by which hypoxia modulates cancer immunity. We provide mechanistic evidence that hypoxia induces T and NK effector cell dysfunction, which occurs through HIF1α-mediated chromatin remodeling resulting in epigenetic suppression of effector gene expression. Further investigation using syngeneic and humanized mouse models demonstrates that therapeutic targeting of HIF1α and its associated epigenetic events effectively reinvigorates the impaired immune effector cell function and enhances anti-PD-1 immunotherapy.

## Results

**Hypoxia is associated with immune escape in breast cancer.** To investigate the role of hypoxia in regulating TNBC immunity, we first sought to characterize the hypoxia gene signature expression in relation to antitumor immune gene activity in TNBC. To this end, data analysis of The Cancer Genome Atlas (TCGA) dataset[11] identified that hypoxic gene signature, which includes commonly documented hypoxia-responsive genes[26], was inversely correlated with immune effector genes (*IFNG, TNF, GZMB, PRF1*), IFNγ-responsive genes, as well as genes associated with tumor-infiltrating lymphocytes (TILs) in TNBC ($n = 99$) (also referred as a basal subtype of breast cancer) (Fig. 1a). Further analysis of other breast cancer subtypes revealed that the inverse correlation between hypoxia gene signature and immune gene activity was also observed in HER2[+] breast cancer but not in luminal breast cancers, which were less aggressive and less hypoxic (Fig. 1b). This observation supports that hypoxia is negatively associated with immune activity in more aggressive breast cancer such as TNBC and HER2[+27]. Compared to luminal breast cancers and

HER2[+] breast cancer, which benefit from hormone or targeted therapies, TNBC remains a significant clinical challenge, which prompted us to focus on TNBC in this study. We further employed immunofluorescent staining to evaluate the spatial relationship between tumor hypoxia and CD8[+] T cell infiltration and IFNγ expression in TNBC. Intratumoral analysis showed that tumor areas with high levels of hypoxia, as indicated by staining of HIF1α, displayed markedly reduced CD8[+] T cell infiltration as well as IFNγ expression compared to low hypoxic areas (Fig. 1c, d). Consistent with the gene signature analysis in TCGA, an inverse correlation between HIF1α and IFNγ[+]/ CD8[+] T cells in TNBC was confirmed (Fig. 1e). We also validated the above findings using a 4T1 murine breast tumor model, which is closely related to human TNBC tumorigenesis and is highly hypoxic[28,29]. In this model, mice bearing the 4T1 xenograft tumors were injected with a chemical substance, pimonidazole (PIM) to probe the tumor hypoxia state[16,19], which allows both immuno-fluorescent and flow cytometry analysis in collected tumors. Tumor areas showing higher levels of PIM exhibited minimum staining of IFNγ and CD8 signals, while the tumor areas showing lower levels of PIM were enriched with IFNγ and CD8 signals (Fig. 1f). Flow cytometry analysis also confirmed that intratumor T cells with higher levels of PIM exhibited lower numbers of CD8[+] T cells and reduced IFNγ expression in CD8[+] T cells (Fig. 1g).

Furthermore, Kaplan–Meier analysis using public breast cancer meta-analysis database[11,30] reveals that TNBC patients with higher hypoxia gene signature are associated with worse disease outcomes, including both overall survival (OS) and distant metastasis-free survival (DMFS), compared to those with lower expression of hypoxia gene signature (Fig. 1h). Conversely, TNBC patients with higher immune effector gene signatures are associated with prolonged survivals (Fig. 1h). A similar result was also found in HER2[+] breast cancer but not in luminal breast cancers (Fig. S1). Together, these findings support that hypoxia has a role in immune exclusion in TNBC and is associated with poor disease outcomes.

**Hypoxia induces T cell dysfunction, which upon further antigen stimulation, drives towards an exhaustion-like state.** To validate and investigate the above findings in clinical samples, we sought to set up an in vitro system to model the effect of hypoxia on human immune cells in TME. We first established a protocol to differentiate naïve human T cells from peripheral blood mononuclear cells (PBMC) of healthy donors into antigen-specific memory and cytotoxic effector cells. We isolated HLA-A2[+] human PBMC from healthy donors, followed by T cell activation and differentiation into dendritic cells (DC) in a procedure depicted in Fig. 2a. DCs were further treated with lysates of HLA-A2[+] TNBC cell line BT549 to allow antigen exposure. After that, tumor antigen-primed DCs were used to stimulate antigen-specific T cell activation. For continuous antigen stimulation, DC-activated T cells were further cocultured with BT549 to achieve further activation and differentiation towards effector function. The flow cytometric analysis shows that T cells persistently activated by tumor antigens using this protocol displayed a progressive increase in CD45RO[−]CD62L[−] effector T cell (Teff) and CD45RO[+]CD62L[−] effector memory T cell (Tem) populations. Meanwhile, decreases in CD45RO[−]CD62L[+] naïve (Tn) and CD45RO[+]CD62L[+] central memory populations (Tcm) were observed when compared with T cells in the resting stage (Fig. 2b). Here, we showed that DC-stimulated, antigen-specific T cells developed in this protocol significantly increased their ability to kill TNBC cells than non-DC-stimulated T cells (Fig. S2a). Therefore, we showed that this protocol could efficiently

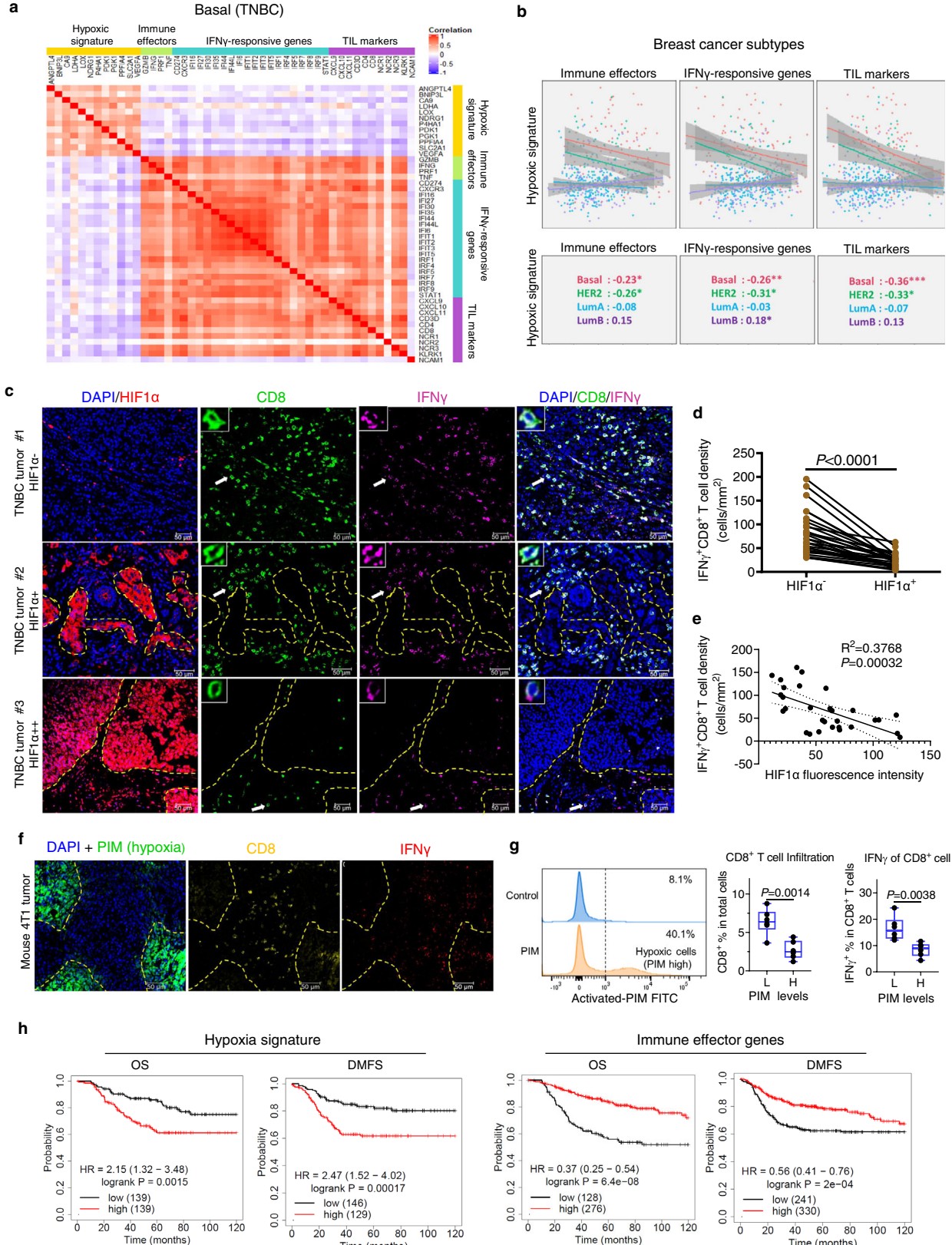

differentiate the peripheral naive T cells into antigen-specific cytotoxic effector and memory T cells.

To investigate the effect of hypoxia on T activation and differentiation process, we further subjected the above T and TNBC coculture to hypoxia (1% O$_2$) (Fig. 2c). Hypoxia did not seem to affect the CD4$^+$ and CD8$^+$ T cell compositions (Fig.

S2b), nor did it affect the overall T cell differentiation status, though it slightly decreased the Tem population and increased Teff population (Fig. S2d). To interrogate the molecular changes induced by hypoxia in differentiated T cells, T cells from three donors cultured in normoxia and hypoxia in the above conditions were subject to RNA-seq analysis (Fig. 2c). The data analysis

**Fig. 1 Hypoxia is associated with immune exclusion in human and mouse triple-negative breast cancer (TNBC). a** Heatmap showing Pearson's correlation between hypoxic signature genes expression and immune-related genes expression in basal TNBC samples ($n = 98$) in TCGA dataset. **b** Scatter plots (upper panel) and Pearson's correlation coefficients (lower panel) showing the expression of hypoxic gene signatures and immune-related genes in breast cancers in TCGA dataset (Basal, $n = 98$; HER2, $n = 58$; Luminal A, $n = 231$; Luminal B, $n = 129$). Regression lines with a 95% confidence interval (gray fill) are shown in the scatter plots. **c** Images of fluorescent staining of human TNBC samples. Scale bar, 50 μm. Data were representative of 30 independent experiments. **d** Quantification of infiltrating IFNγ$^+$ CD8$^+$ T cell number in HIF1α$^-$ and HIF1α$^+$ regions of human TNBC sample ($n = 30$). $P$ values were determined with paired two-tailed $t$-test. **e** Correlation between infiltrating IFNγ$^+$ CD8$^+$ T cell count and HIF1α fluorescent intensity in human TNBC samples ($n = 30$). The simple linear regression $R^2$ and $P$ values (two-tailed) are calculated. Dot plot is shown with regression line and 95% confidence interval. **f** Representative images of fluorescent staining of mouse 4T1 tumor samples. Scale bar, 50 μm. Data represents three independent experiments. **g** Flow cytometry (left panel) demonstrating the gating strategy of activated-PIM high (H) and activated-PIM low (L) populations in living cells dissociated from 4T1 tumors. The CD8$^+$ T cell percentage and IFNγ expression in CD8$^+$ T cells was quantified (right panel, $n = 6$). Data were presented as box and whiskers, with median value and whiskers of minimum and maximum values. $P$ values were determined with an unpaired two-tailed $t$-test. **h** Kaplan–Meier overall survival (OS) and distant metastasis-free survival (DMFS) analysis of the indicated gene signatures in TNBC patients. The publicly available data used in Fig. 1a, b are available in the TCGA database under accession code BRCA.exp.547.med.txt [https://gdc.cancer.gov/about-data/publications/brca_2012]. The publicly available data used in **h** are available in the KM-Plotter-Breast Cancer [https://kmplot.com/analysis/index.php?p=service&cancer=breast]. For the remaining data, source data are provided in Source Data file.

showed that hypoxia treatment resulted in substantial transcriptional changes in T cells. These included marked downregulation of a cohort of cytokines expression, one of the critical hallmark features of T cell exhaustion[25], accompanied by upregulation of genes enriched in the hypoxia pathway (Fig. 2d, e; twofold change, $p < 0.01$ as cutoffs). Among them, *IFNG* (IFNγ) was one of the top genes downregulated in hypoxia and was consistently downregulated in all three donors. *TNF* (TNFα) and *GZMB* (granzyme B) were also downregulated by hypoxia in two of the three donors (1.7–2.4 fold), though they did not meet the cutoff criteria.

We next used flow cytometry to profile the protein expression of immune effector and immune checkpoint proteins in both T cells alone (monoculture) or T cells coculture with TNBC cell line BT549 under normoxia and hypoxia. The results showed that hypoxia caused marked downregulations of IFNγ, TNFα, and granzyme B in CD8$^+$ T cells in both monoculture and coculture conditions (Fig. 2f). Interestingly, hypoxia significantly induced TIM-3 and TIGIT protein expression, though it did not affect the classic checkpoint molecules PD-1 and CTLA-4 (Fig. 2f). Compared to hypoxia treatment of T cell alone, hypoxia together with coculture with TNBC cells resulted in further upregulation of TIM-3 and TIGIT expression in CD8$^+$ and CD4$^+$cells (Fig. 2f and Fig. S2c). TIM-3 and TIGIT are co-inhibitory molecules whose co-expression with PD-1 or/ and CTLA-4 are one of the characteristics of T cell exhaustion commonly found in TILs[19,31]. Indeed, hypoxia plus coculture resulted in significant upregulation of TIM-3$^+$/PD-1$^+$ double-positive population (30–40%) in CD8$^+$ T cells compared to T cells in the resting stage (10–20%) (Fig. 2g, h). However, the hypoxia did not alter the expression of Ki67 (Fig. 2i), indicating that the above hypoxic effect is not due to a change in cell proliferation. Therefore, the data suggest that hypoxia not only induces immune effector downregulation it can also work together with tumor antigen stimulation to further enhance the exhaustion signals, creating an exhaustion-like state. Of notice, hypoxia-induced suppression of immune effectors occurred across all the subsets of CD8$^+$ cells, despite more obvious in effector subsets (both Tem and Teff) (Fig. S2e), indicating that hypoxia can induce effector suppression regardless of T cell differentiation.

Furthermore, in T cells stimulated with CD3/CD28 antibodies in the absence of tumor antigen, hypoxia remains capable of suppressing the effector expression of CD8$^+$ and CD4$^+$ T cells (Fig. S2f). Hypoxia-induced suppression of effector expression and upregulation of TIM-3 was also observed in NK cells isolated from PBMC (Fig. S2g, h), though the effect on TIGIT was not so obvious upon coculture with TNBC (Fig. S2h). Thus, we have

demonstrated that hypoxia is sufficient to suppress the immune effector gene expression in T and NK cells. This effect is most likely a direct effect rather than a secondary effect of immune cell exhaustion, which raised our interest in investigating the corresponding mechanism.

**Hypoxia induces chromatin remodeling to suppress immune effector gene expression in T and NK cells.** Epigenetics plays a vital role in immune cell dysfunction and exhaustion that can limit response to immunotherapy[32–36]. We next performed an epigenetic compound screening to investigate whether hypoxia engages an epigenetic mechanism to suppress the effector expression. To do this, we cultured anti-CD3/anti-CD28-activated human T cells and anti-CD335/anti-CD2-activated NK cells under hypoxia and treated the cells with epigenetic compounds for 48 h. RT-PCR assessment of *IFNG* expression was used as a readout to determine the capacity of the epigenetic drugs to restore *IFNG* expression in hypoxia. Among the drugs tested that target different epigenetic modifications, inhibitors of histone deacetylase (HDAC) and Enhancer of zeste homolog 2 (EZH2)/PRC2, and to a lesser extent, DNMT inhibitor 5-azacytidine (AZA), were found to consistently induce the expression of *IFNG* in both T and NK cells (Fig. 3a). We decided to focus on HDAC and EZH2 inhibitors as DNMT inhibitors have been previously shown to induce *IFNG* expression, though not in hypoxia[37]. Further verification by flow cytometry has demonstrated that HDAC inhibitor Entinostat (ENT) and EZH2 inhibitor EPZ6438 (EPZ), both of which are clinically approved drugs, were able to rescue the IFNγ, TNFα and granzyme B expression in hypoxic CD8$^+$, CD4$^+$ T cells and NK cells (Fig. 3b, c and Fig. S3a, b).

Given the above findings, we next performed a chromatin immunoprecipitation (ChIP) assay to examine if hypoxia-induced enrichments of HDACs and PRC2 at the effector gene promoters in the T and NK cells. Scanning the promoter regions across −1.5 to 1 Kb of TSS of *IFNG* and *TNF* showed that although with minimum enrichments of HDACs in normoxia, hypoxia treatment resulted in higher HDAC1 enrichment at both *IFNG* and *TNF*, though without affecting HDAC2 and HDAC3 (Fig. 3d and Fig. S3c). Interestingly, EZH2 and its PRC2 partner SUZ12 already showed a significant enrichment at *IFNG* in normoxia, indicating a pre-existing PRC2 complex at *IFNG*, and hypoxia did not further increase their enrichments (Fig. 3d). Corresponding to the increased HDAC1 enrichment in hypoxia, H3K27ac at *IFNG* and *TNF* showed decreased enrichment in hypoxia while H3K27me3 enrichment was increased (Fig. 3e and Fig. S3d). Similar results

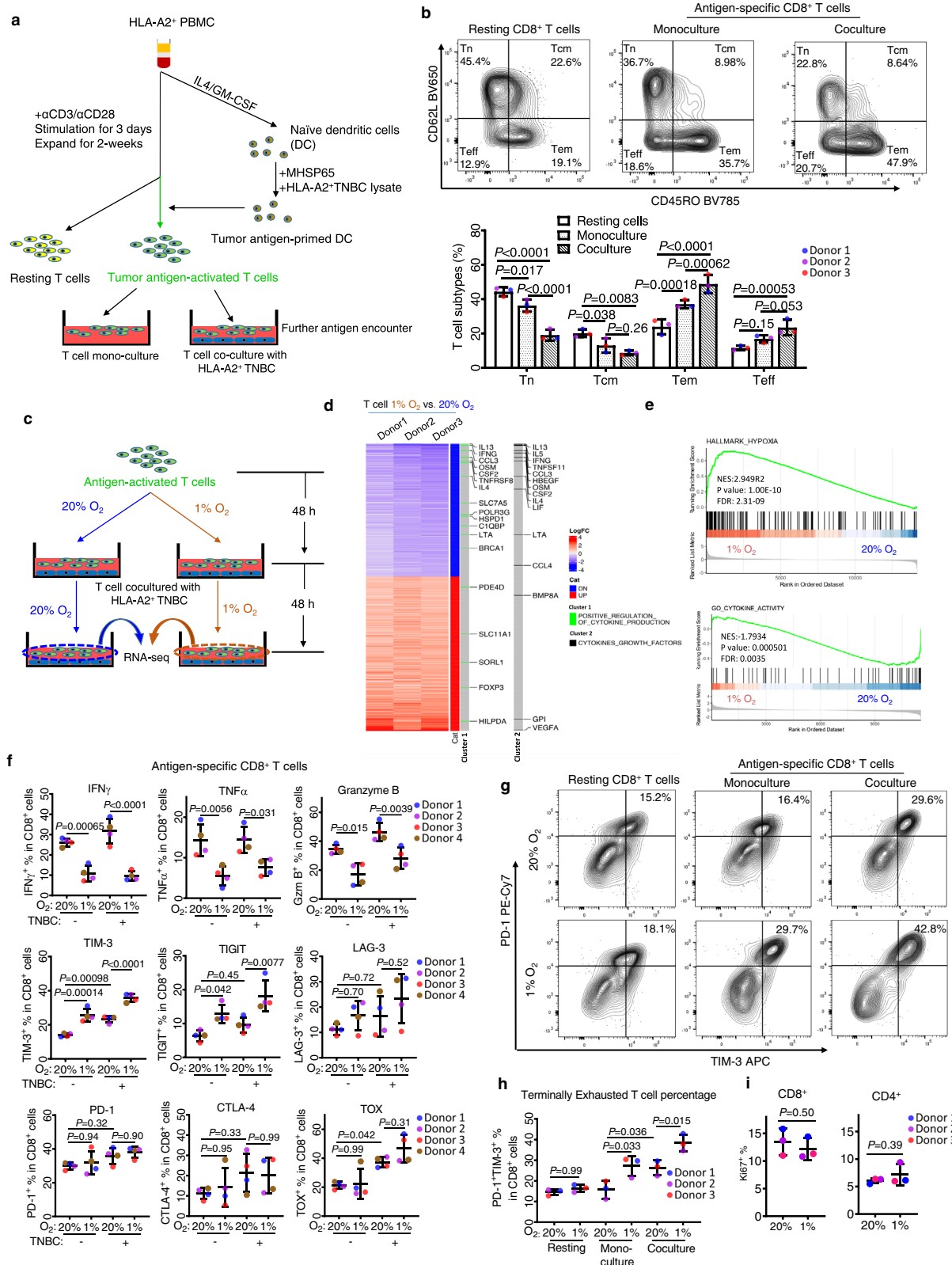

were also seen in NK cells (Fig. S3e, f). It has been shown that HDACs can work in coordination with the PRC2 complex in histone modulations, and H3K27ac and H3K27me3 antagonize each other to maintain a dynamic equilibrium of chromatin activity[38,39]. Our observation is consistent with notation and suggests that in normoxia the higher expression of effector genes are associated with abundant H3K27ac that counteracts

the H3K27me3 despite the presence of PRC2. Upon hypoxia, increased HDAC1 removes the H3K27ac, which increased H3K27me3 by the pre-existing PRC2 and accordingly reduced effector expression. Consistent with this hypothesis, we see that HDAC inhibition by HDAC inhibitor ENT or EZH2 inhibition by EPZ reversed the hypoxia-induced changes in H3K27ac and H3K27me3 at *IFNG* (Fig. 3f).

**Fig. 2 Hypoxia induces dysfunction and terminal exhaustion of human T cells. a** Schematic graph demonstrating the coculture model. **b** Representative flow cytograms (upper panel) gated from human pan-T cell culture and quantification (lower panel, $n = 3$) of differentiated CD8$^+$ T cell subtypes: Tn (naïve T cells), Tcm (central memory T cells), Tem (effector memory T cells), Teff (effector T cells). **c** Schematic graph demonstrating the normoxia (20% O$_2$) and hypoxia (1% O$_2$) culture condition of T cells coculturing with human TNBC cell line. **d** Heatmap of the differentially expressed genes (DEGs) in hypoxic cultured human T cells compared to normoxia group. DEGs were identified in edgeR (|logFC| > 1, adjusted $P < 0.01$). $P$ values were adjusted using Benjamini–Hochberg method in edgeR. DEGs identified in the indicated GO gene clusters are marked in the heatmap. **e** GSEA analysis of human T cells in hypoxic versus normoxic conditions. Analysis was based on ranked logFC from edgeR. FDR and adjusted $p$ value are shown in the graph. $P$ values were adjusted using Benjamini–Hochberg method in GSEA analysis. **f** Flow cytometry quantifications of immune effector molecules and exhaustion markers in CD8$^+$ T cells gated from human pan-T cells cultured under the indicated conditions ($n = 4$). **g** Representative flow cytograms of PD-1 and TIM-3 expression in CD8$^+$ T cells gated from human pan-T cells culture. **h** Flow cytometric quantification of terminally exhausted T cells (PD-1$^+$ TIM-3$^+$) in CD8$^+$ T cells gated from human pan-T cells culture ($n = 3$). **i** Flow cytometric quantification of proliferating cells (Ki76$^+$) in CD8$^+$ and CD4$^+$ T cells gated from human T cells cocultured with TNBC ($n = 3$). All flow cytometry data (**b**, **f**, **h**, and **i**) are presented as the mean ± SD of samples from three to four donors. For all flow cytometry data, $P$ values were determined by one-way ANOVA (**f**, **h**) or two-way ANOVA (**b**) with Turkey's test, or paired two-tailed $t$-test (**i**). Raw RNA-seq data **i**s available in the GEO database with accession number GSE179885. For the remaining data, source data are provided in Source Data file.

Further validation using CRISPR-mediated depletion of HDAC1 or EZH2 recapitulated the effect of ENT or EPZ and restored effector gene expression in T and NK cells (Fig. S4a, b, Fig. 3g, and Fig. S4c). Of note, hypoxia did not change the bulk expression levels of HDACs, EZH2, SUZ12, and EED (Fig. S4d), nor the global levels of H3K27me3 and H3K27ac in T and NK cells (Fig. S4d). These findings showed that the hypoxia-induced epigenetic inactivation of effector genes is not due to overall changes in epigenetic enzymes or histone modifications. Therefore, hypoxia triggered an epigenetic remodeling and HDAC1/PRC2 concurrent dependency, resulting in effector gene suppression.

**Hypoxia-induced epigenetic inactivation of effector genes is HIF1α-dependent.** Hypoxia induces both HIFs-dependent and HIFs-independent molecular changes[40]. To investigate whether HIFs are involved in hypoxia-induced immune effector dysfunction, we performed ChIP analyses of HIF1α and HIF2α at *IFNG* and *TNF* promoters. The results show that hypoxia treatment of both T and NK cells induced a marked increase in HIF1α, but not HIF2α, enrichments at *IFNG* and *TNF* in regions identical to HDAC1 binding (Fig. 4a and Fig. S5a, b). This finding suggests that hypoxia promotes co-enrichments of HIF1α and HDAC1 at *IFNG* and *TNF*. We next performed a co-immunoprecipitation assay to investigate whether this co-enrichment is due to the physical interaction between HIF1α and HDAC1. We found that HIF1α co-immunoprecipitated with HDAC1, suggesting that HDAC1 might form a complex with HIF1α under hypoxia. HDAC1, but not HIF1α, was also found in SUZ12 pulldown (Fig. 4b). These findings suggest that HDAC1 interacts with HIF1α and PRC2 after hypoxia to form two distinct repressor complexes that co-occupy the effector gene promoters. HIF1α knockdown in T cells abolished hypoxia-induced HDAC1 enrichment at effector genes (Fig. 4c, d and Fig. S5d), reversed the corresponding changes in H3K27ac and H3K27me3 (Fig. 4e and Fig. S5e), and restored *IFNG* expression in hypoxia (Fig. 4f). Likewise, pharmacologic depletion of HIF1α by small molecules PX478 and Digoxin[41,42] also produced similar results in T and NK cells (Fig. 4g, h and Fig. S5f, g, h). These data support that HIF1a induction by hypoxia increased the HDAC1 recruitment to the effector gene promoter to enable their downregulation. To further validate the role of HIF1α in suppressing effector expression, we ectopically overexpressed *HIF1A* in T cells under normoxia and T cells with *HIF1A* KD under hypoxia (Fig. S6a). Like hypoxia treatment, *HIF1A* overexpression was sufficient to decrease effector expression in T cells under normoxia; it also abolished effector induction in T cells with *HIF1A* KD under hypoxia (Fig. S6b). Taken together, we conclude that hypoxia

resulted in HIF1α-dependent epigenetic reprogramming through HDAC1 and PRC2-mediated histone modifications, enforcing the transcriptional suppression of effector genes in human T and NK cells. We also found that neither hypoxic culture nor the drug treatment significantly affects the viability of T cells within a time course of 6 days (Fig. S6c). Interestingly, HIF1α knockdown or inhibitors of HIF1α, HDAC, or EZH2 did not affect hypoxia-induced TIM-3 and TIGIT expressions (Fig. S6d, e), nor did they affect the percentage of PD-1$^+$TIM-3$^+$ double-positive populations in CD8 T cells under hypoxia (Fig. S6f), but increased IFNγ levels in PD-1$^-$TIM-3$^-$, PD-1$^+$TIM-3$^-$ and PD-1$^+$TIM-3$^+$ populations of CD8 T cells under hypoxia (Fig. S6g). The data indicate that HIF1α and its associated epigenetic machinery act directly on the expression of effector cytokines, regardless of T cells' differentiation and inhibitory receptor expression patterns. Therefore, hypoxia drives the T cell to an exhausted-like state through both HIF1α-dependent and HIF1α-independent mechanisms. Nevertheless, we show that the HIF1α-dependent epigenetic mechanism is an important driver event inducing effector dysfunction and therapeutic targeting of this mechanism appeared to be sufficient to rescue the impaired immune effector expression.

**Hypoxia impairs the immune cell cytotoxicity, reduces interferon signaling response, and confers resistance to anti-PD-1 blockade.** To investigate whether hypoxia-induced suppression of effector expression in T and NK cells leads to impaired cytotoxicity, we performed T or NK cell-mediated cytotoxicity towards TNBC cells BT549 and MDA-MB-231 expressing a luciferase reporter. As anticipated, we observed that hypoxia treatment of T or NK cells resulted in a reduced capacity in killing TNBC cells, as determined by measuring the luciferase activity of the coculture (Fig. 5a and Fig. S7a).

IFNγ is the most downregulated effector by hypoxia in T and NK cells, and interferon signaling is of central importance to antitumor immune response[43]. We next evaluated the effect of hypoxia on T or NK–elicited interferon signaling hallmark proteins STAT1, PD-L1, and interferon regulatory factor 1 (IRF1). In both TNBC cell lines, hypoxia attenuated immune cells-elicited interferon signaling, as evidenced by both Western blotting (Fig. 5b) and RT-qPCR (Fig. S7c, d). Furthermore, pretreatment of T cell or NK cells with inhibitors of HIF1α (Digoxin and PX478), EZH2 inhibitors (EPZ and GSK126), and HDAC inhibitors (ENT and SAHA) were able to rescue the impaired killing of TNBC cells by immune cells in hypoxia (Fig. 5c), with the corresponding restoration of interferon signaling response (Fig. 5d and Fig. S7e, f). By contrast, in the absence of immune cells, neither hypoxia nor the above drugs

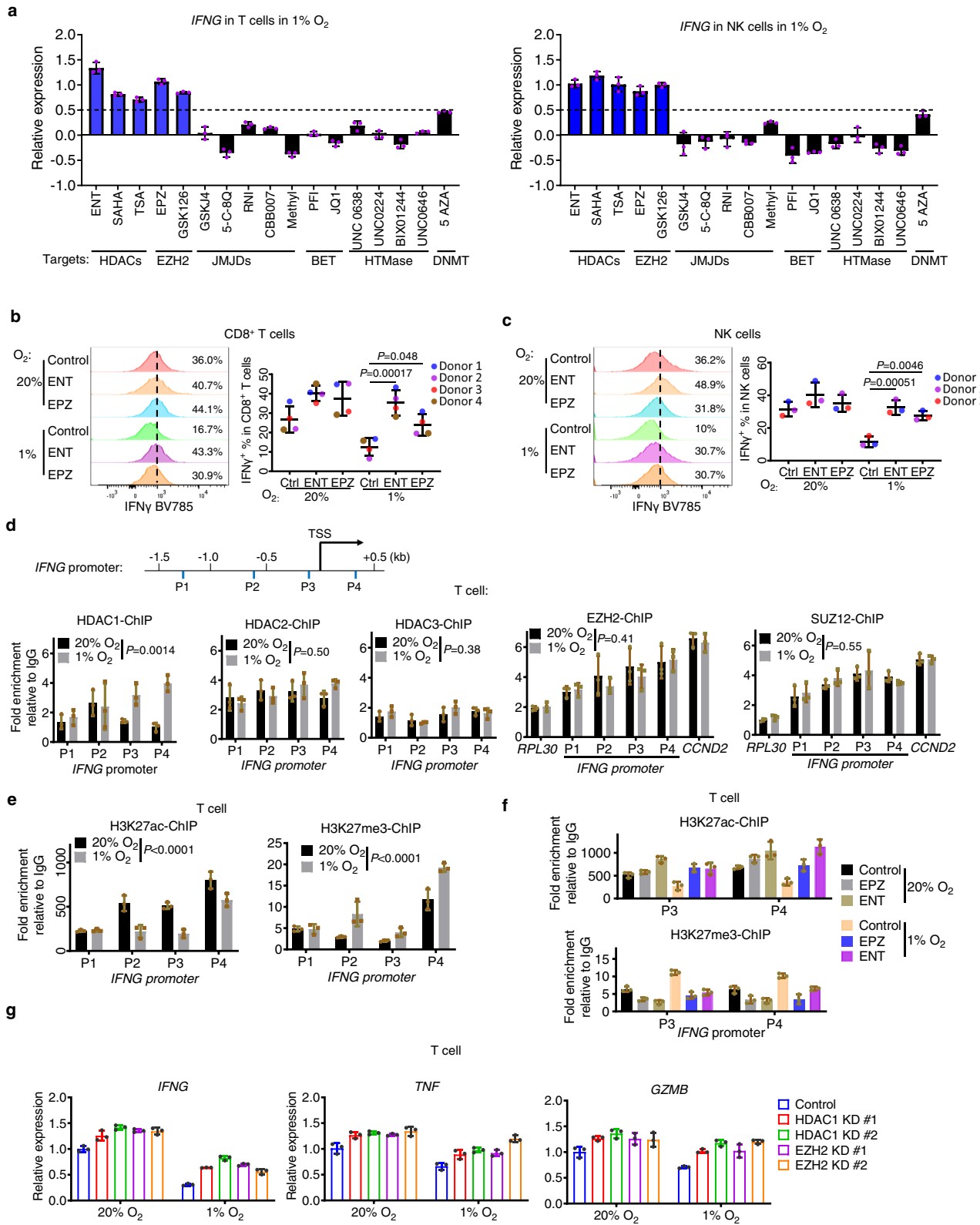

altered the *IRF1* and *PD-L1* expression and the viability of the two TNBC cell lines (Fig. S7g, h). These findings consolidated that hypoxia acts on the immune cells to weaken the interferon signaling in TNBC cells rather than directly influence TNBC cells.

Inhibition of IFNγ production and interferon response including PD-L1 is expected to induce resistance to the PD-1/PD-L1 blockade. Moreover, although inhibition of HIF1α,

HDAC1, or EZH2 in immune cells can restore immune cytotoxicity, it also paradoxically induces PD-L1 and PD-L2 expression in tumor cells that in turn could weaken anticancer immunity. Therefore, we envision that PD-1/PD-L1 blockade plus inhibitors of HIF1α, HDAC1, or EZH2 appears to be a rational combination that could overcome the obstacle of immunotherapy. Indeed, we found that although anti-PD-1

**Fig. 3 Hypoxia induces epigenetic inactivation of immune effector expression. a** RT-qPCR analysis assessing *IFNG* expression in T/NK cells in an epigenetic-drug screening. Both T cells and NK cells were cultured under 1% $O_2$ with indicated treatments. Data were presented as the log2 fold change of *IFNG* mRNA level normalized to vehicle control, mean ± SD of technical triplicates, representative of two independent experiments ($n = 2$). **b**, **c** Representative histograms (left panel) and flow cytometric quantifications (right panel) of IFNγ expression in human CD8[+] T cells (**b** $n = 4$) and NK cells (**c** $n = 3$) with indicated treatments. Quantification data were presented as the mean ± SD of samples from three to four donors. *P* values were determined by two-way ANOVA with Turkey's test. **d** ChIP-qPCR analysis of HDAC1, HDAC2, HDAC3, EZH2, and SUZ12 occupancy on *IFNG* promoter of human T cells. Four primers were designed to span the promoters of *IFNG*, with P1 at −1448 to −1354b, P2 at −707 to −628b, P3 at −257 to −171b, P4 at +350 to +461b, relative to TSS. For ChIP analysis of EZH2 and SUZ12 occupancy, *RPL30* serves as the negative control and *CCND2* as the positive control. **e**, **f** ChIP-qPCR analysis of H3K27ac and H3K27me3 enrichment on *IFNG* promoter of human T cells under indicated conditions. All ChIP-qPCR data (**d**–**f**) are presented as fold enrichment relative to IgG and expressed as mean ± SD of technical triplicates, representative of two independent experiments ($n = 2$). For ChIP-qPCR data of **d**, **e**, statistics were performed to analyze bindings of indicated markers across different sites in *IFNG* promoter (*RPL30* and *CCND2* excluded) between hypoxia and normoxia. *P* values were determined by two-way ANOVA analysis. **g** RT-qPCR analysis of human T cell with indicated gene knockdown. Data were presented as the fold change of mRNA level normalized to the control group under normoxia (1% O2), mean ± SD of technical triplicates, representative of two independent experiments ($n = 2$). Source data are provided as a source data file.

antibody can induce T cell killing of BT549 and MDA-MB-231 cells in normoxia, it failed to do so in hypoxia, verifying that hypoxia conferred resistance to PD-1 blockade. Indeed, hypoxia-induced resistance to anti-PD-1 was reversed by PX478, ENT, and EPZ, as well as the knockdown of HIF1α, HDAC1, or EZH2 (Fig. 5e). Consistently, flow cytometry analysis also showed that hypoxia blunted the induction of effector expression by anti-PD-1 in CD8[+] T cells, which can be rescued by treatment with PX478, ENT, or EPZ (Fig. 5f). Collectively, these in vitro findings validated the role of hypoxia and HIF1α-associated epigenetic events in impairing immune cytotoxicity, interferon signaling, and therapeutic response to immunotherapy.

**Targeting HIF1α and associated epigenetic dependency overcomes resistance to PD-1 blockade in both syngeneic and humanized mouse models.** To interrogate the efficacy of HIF1α/HDAC1/EZH2 inhibitors in vivo, we used the immune-competent syngeneic 4T1 murine breast tumor model that closely mimics human TNBC breast cancer tumorigenesis[28]. In addition, the 4T1 xenograft tumor is also known to be highly hypoxic and resistant to PD-1 blockade treatment[44,45]. To enable the in vivo validation using this mouse model, we first sought to confirm if hypoxia and continued antigen stimulation of mouse T cells could also lead to mouse T cell exhaustion, as seen in human T cells. Indeed, hypoxia treatment of the ex vivo-expanded mouse T cells isolated from the spleen of a 4T1-bearing mouse resulted in downregulations of IFNγ, TNFα, and granzyme B but upregulations of TIM-3 and TIGIT (Fig. S8a, b, and c). PX478, ENT, and EPZ treatment restored the effector expression in hypoxia, and as in human T cells, they did not affect TIM-3 expression (Fig. S8d). Hypoxia also weakened the mouse T cell cytotoxicity and anti-PD-1 response towards 4T1, and inhibitors of HIF1α, HDAC1, and EZH2 restored the anti-PD-1 antibody treatment response (Fig. S8e). Thus, these in vitro studies in mouse T cells recapitulated the results seen in human T cells.

Further studies in vivo using 4T1-engrafted BALB/c mice showed that anti-PD-1 treatment alone had minimum effect on tumor growth, but a combination of anti-PD-1 with PX478 or ENT could induce substantial tumor regression (Fig. 6a and Fig. S9a). The combination treatments also resulted in robust suppression of lung metastasis compared to single treatments (Fig. 6b and Fig. S9d), which were translated to significant survival benefits (Fig. 6c). EPZ combined with anti-PD-1 also improved the effect on tumor growth and metastasis, though the effects were less potent than PX478 and ENT (Fig. S9b, c), probably due to poor PD/PK issue because EPZ couldn't achieve a satisfying solubility in different formulations in our hands. Moreover, in mice with depletion of T or NK cells, the efficacy of combination treatment was largely abolished (Fig. 6d and

Fig. S9e). T cell depletion also markedly promoted lung metastasis and abolished the drug combination effects on metastasis, though NK depletion only exhibited a modest impact on metastasis (Fig. 6e). Thus, both T and NK cells were required for efficient tumor control induced by the combination treatments, though T cells seemed to be more crucial in controlling the metastasis progression.

We next sought to confirm if the above treatments reversed the immune effector gene expression in intratumoral T and NK cells. Flow cytometry analysis of resected 4T1 tumors collected from a separate experiment showed that the combination treatment resulted in increased expression of IFNγ, TNFα, and granzyme B in intratumoral CD8[+] T and NK cells (Fig. 6f and Fig. S9f), as well as an overall increase in tumor-infiltrating CD8[+] T cells (Fig. 6f). In addition, consistent with the induction of IFNγ, we also observed significantly increased expression of IFNγ responsive PD-L1 and PD-L2 (Fig. 6f) as well as *CXCL9*, *CXCL10*, *CXCL11*, and *IRF1* in tumors treated with combination treatments (Fig. S9g). By contrast, the combination treatments did not induce the IFNγ expression in T and NK cells isolated from the mouse spleen or peripheral blood of the same mice (Fig. S9h). These analyses demonstrated the specificity of the combinational therapies towards intratumoral immune cells, which converted a cold tumor TME towards a hot one that is permissive to immune checkpoint blockade.

To confirm the effect of hypoxia as well as the above treatments on the 4T1 tumor T cell infiltration, we applied pimonidazole (PIM), to probe the tumor hypoxia[16,19]. Flow cytometry analysis of PIM and HIF1α demonstrated a positive correlation between HIF1α expression and PIM and treatment with PX478 or PX478/anti-PD-1 markedly depleted HIF1α expression (Fig. 6g). Immunofluorescence imaging of paraformaldehyde-fixed 4T1 tumors confirmed that tumor areas with high hypoxia reduced infiltration of CD8[+] T cells, and the combination treatments reversed the hypoxic effect and promoted the effector T cell infiltration into the hypoxic region (Fig. 6h, i). Further flow cytometry analysis of resected 4T1 tumors showed CD8[+] TIL with higher PIM staining displayed lower expressions of IFNγ and TNFα (Fig. 6j). Anti-PD-1 treatment alone failed to restore the IFNγ and TNFα expression on CD8[+] cells, but combinations with ENT or PX478 restored the expression of IFNγ and TNFα in CD8[+] TILs and blunted the hypoxic effect (Fig. 6j). In addition, using HIF1α staining also gave rise to similar results (Fig. 6k). Taken together, we have demonstrated that inhibition of HIF1α and its binding partner HDAC1 can reinvigorate impaired immune cell effector activity in a hypoxic TME, leading to an enhanced response to PD-1 blockade.

We further sought to validate the combination efficacy using a humanized mouse model, in which NOD/SCID/IL2Rg[−/−]

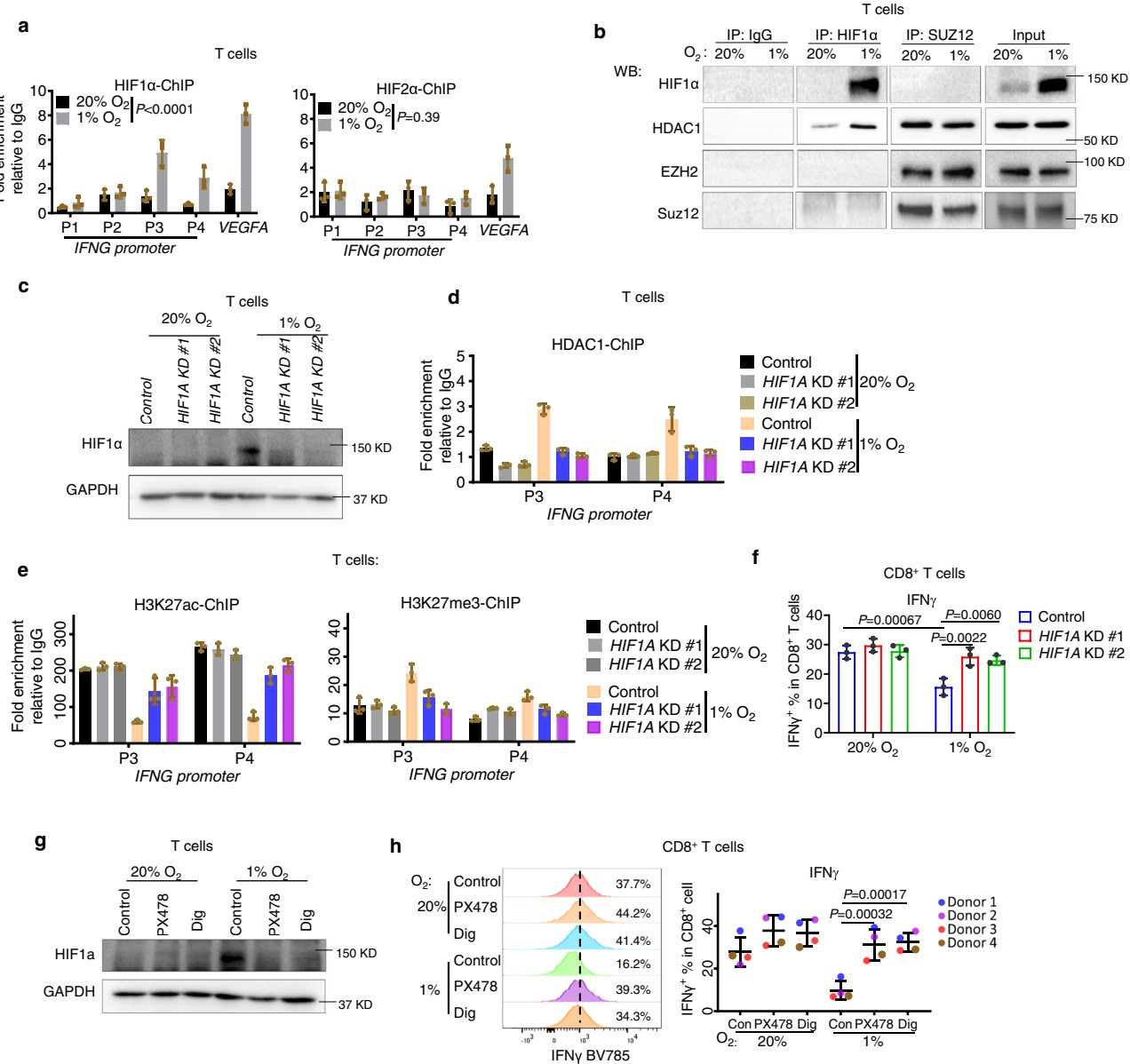

**Fig. 4 Hypoxia-induced epigenetic inactivation of effector expression is HIF1α-dependent. a** ChIP-qPCR analysis of HIF1α and HIF2α occupancy on *IFNG* promoter in human T cells. *VEGFA* served as a positive control. **b** Co-immunoprecipitation shows the physical interaction between HDAC1 and HIF1α, and the interaction between HDAC1 and SUZ12 in human T cells. Data is representative of two independent experiments ($n = 2$). **c** Representative western blot images ($n = 2$) to demonstrate knockdown of HIF1α in human T cells. **d** ChIP-qPCR analysis of HDAC1 occupancy on *IFNG* promoter in human T cells. **e** ChIP-qPCR analysis of H3K27ac and H3K27me3 enrichment on *IFNG* promoter in human T cells with indicated treatments. All ChIP-qPCR data (**a**, **d**, **e**) are presented as fold enrichment relative to IgG and expressed as mean ± SD of technical triplicates, representative of two independent experiments ($n = 2$). For ChIP-qPCR data of **a**, statistics were performed to analyze bindings of indicated markers across different sites in *IFNG* promoter (*VEGFA* excluded) between hypoxia and normoxia. $P$ values were determined by two-way ANOVA analysis. **f** Flow cytometric quantifications of IFNγ in CD8[+] T cells gated from human pan-T cells cultured under the indicated conditions. Data were presented as the mean ± SD of three independent experiments ($n = 3$). $P$ values were determined by one-way ANOVA with Turkey's test. **g** Representative western blot images ($n = 2$) to demonstrate the inhibition of HIF1α level by indicated compounds in human T cells. **h** Representative histograms (left panel) and flow cytometric quantifications (right panel) of IFNγ expression in human CD8[+] T cells with indicated treatments. Quantification data were presented as the mean ± SD of samples from four donors ($n = 4$). $P$ values were determined by two-way ANOVA with Turkey's test. Source data are provided as a source data file.

(NIKO) mice were reconstituted with the human immune system as we previously described[46]. Ten weeks after transplanting CD34[+] human fetal liver cells into 3-day-old NIKO pup, mice were evaluated for human T and NK cells reconstitution (Fig. 7a). We used human TNBC cell line MDA-MB-231-LM2 (LM2) cells for tumor generation, which is an ideal xenograft model for studying tumor growth and lung metastasis[47]. In vitro coculture experiment using LM2 cells showed similar results as MDA-231

cells (Fig. S7b). Successfully transplanted humanized mice were engrafted with human LM2 cells at mammary fat pads to form xenograft tumors and subsequent lung metastasis. In this humanized mouse model, we showed that the anti-PD-1 agent Keytruda alone did not have a pronounced effect, but its combination with PX478 or ENT resulted in marked suppression of tumor growth and lung metastasis (Fig. 7b–d). As a control, LM2-bearing NIKO mice without human immune cells showed a

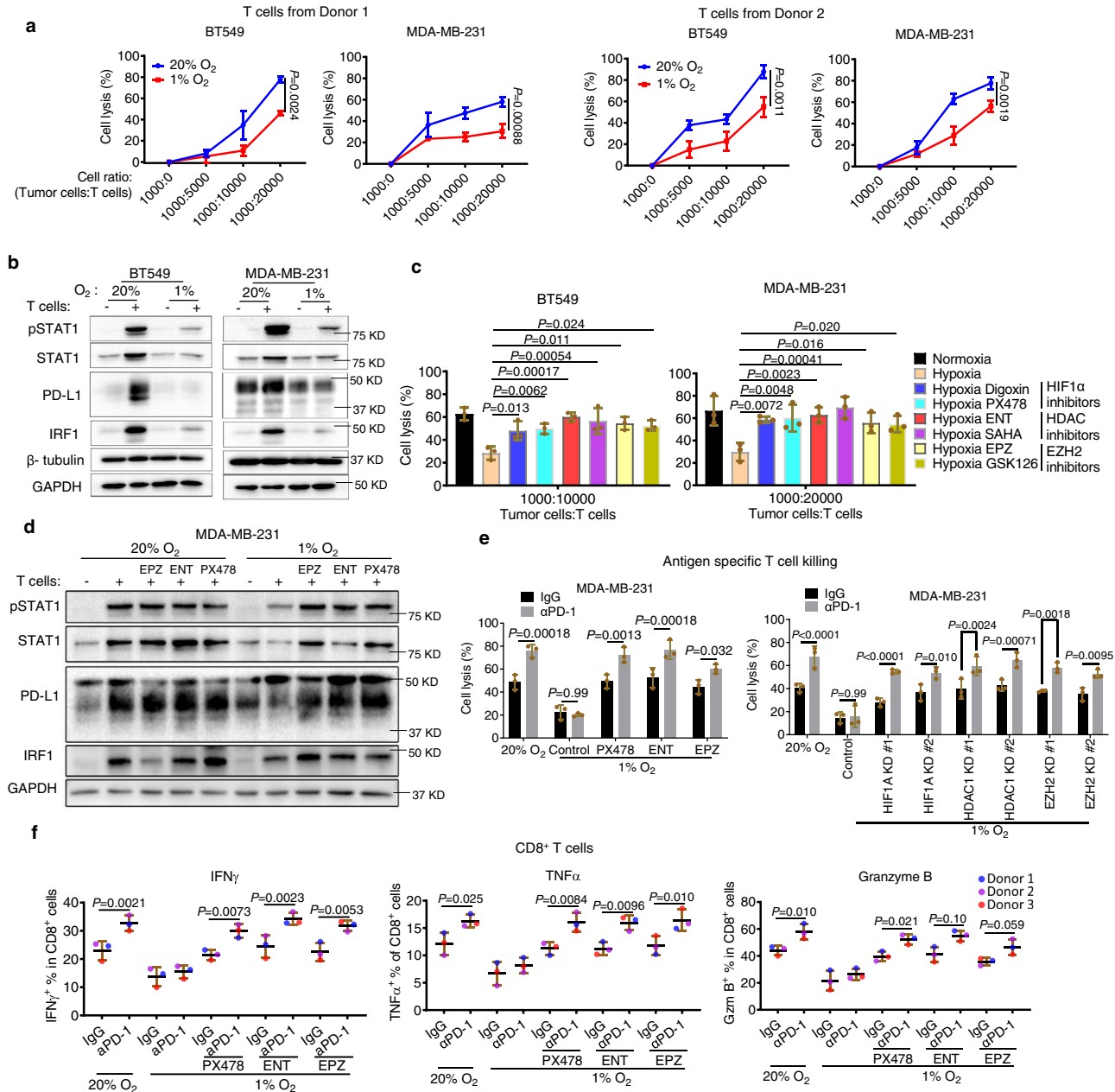

**Fig. 5 Hypoxia dampens immune cytotoxicity, weakens interferon signaling, and induces resistance to PD-1 blockade. a** Cell lysis of TNBC cells cocultured with human T cells from two different healthy donors. Human T cells were stimulated with TNBC cell lysate-primed DC cells. Data were presented as mean ± SD of three independent experiments ($n = 3$). $P$ values were determined by two-way ANOVA. **b** Western blot analysis of IFNγ–regulated proteins in TNBC cells cocultured with human T cells. Data were representative of two independent experiments ($n = 2$). **c** Cell lysis of TNBC cells cocultured with human T cells. Human T cells were stimulated with TNBC cell lysate-primed DC cells and pretreated with indicated compounds. Data presented as mean ± SD of three independent experiments ($n = 3$). $P$ values were determined by one-way ANOVA with Dunnett's test. **d** Western blot analysis of IFNγ–regulated proteins in TNBC cells cocultured with human T cells. Human T cells were stimulated with TNBC cell lysate-primed DC cells and pretreated with indicated compounds. Data were representative of two independent experiments ($n = 2$). **e** Cell lysis of TNBC cells cocultured with human T cells. Data were presented as mean ± SD of three independent experiments ($n = 3$). $P$ values were determined by two-way ANOVA with Dunnett's test. **f** Flow cytometric quantifications of immune effector molecules in human CD8+ T cells cultured under the indicated conditions. Data were presented as the mean ± SD of samples from three donors ($n = 3$). $P$ values were determined by two-way ANOVA with Turkey's test. Source data are provided as a source data file.

much-weakened response to the combination treatments (Fig. 7b–d), validating the utility of the humanized mouse in the evaluation of immunotherapy and related combination treatments.

Similar to the above murine findings, the combination therapies increased human IFNγ, TNFα, and granzyme B production in intratumoral human CD8+ and NK cells compared

to Keytruda treatment alone (Fig. 7e). They also increased expressions of PD-L1 and PD-L2 in tumors that indicate a more active immune environment (Fig. 7f). These findings from both syngeneic and humanized mouse models indicated the potential clinical utility of the proposed combination treatments to reinvigorate the immune dysfunction and sensitize PD-1 blockade in hypoxic TNBC.

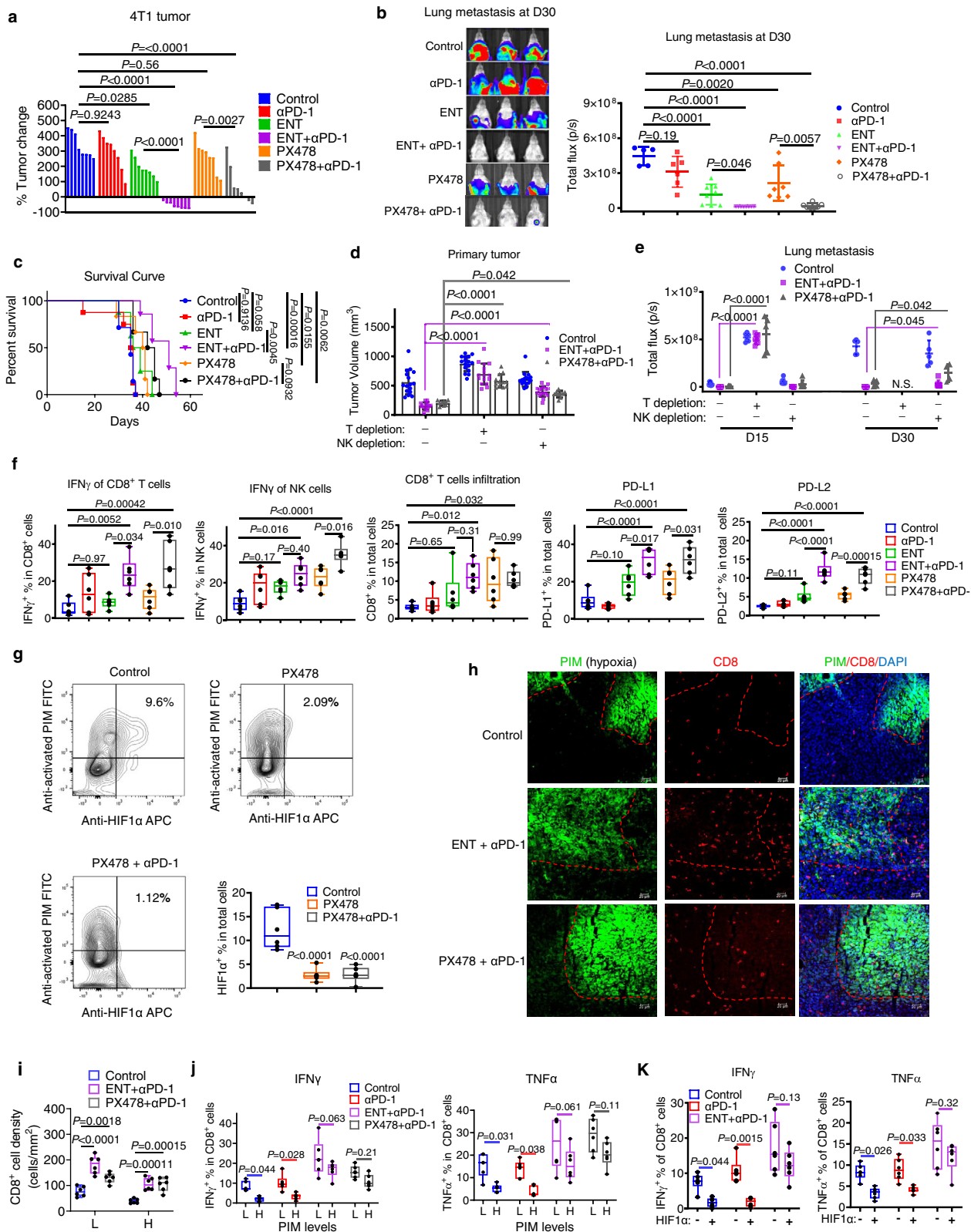

## Discussion

Hypoxia plays a crucial role in tumor growth, metastasis, and drug resistance in solid tumors[8,9], but its role in modulating cancer immunity and the corresponding mechanism remains unclear. In the present study, we demonstrate that hypoxia suppresses immune effector activity through epigenetic reprogramming, which, together with continuous antigen stimulation, is crucial to developing immune exhaustion and resistance to immunotherapy. These findings were supported by in vivo studies using both syngeneic and humanized models in which targeting hypoxia-activated epigenetic machinery was found to sensitize TNBC to anti-PD-1 therapy, thus providing

**Fig. 6 Pharmacological inhibition of HIF1α and associated epigenetic events sensitizes PD-1 blockade in a syngeneic mouse model. a** Change of 4T1 tumor volume from baseline in BALB/c mice at Day 16 of drug treatments. $N = 8$. **b** Lung metastasis of BALB/c mice bearing 4T1 at Day 30. Left panel, representative bioluminescence images. Right panel, quantification of lung metastasis. Control, $n = 5$; αPD-1, $n = 6$; $n = 8$ for ENT and ENT + αPD-1; $n = 7$ for PX478 and PX478 + αPD-1. **c** Kaplan–Meier survival curve for 4T1-bearing mice. Control, $n = 8$; αPD-1, $n = 7$; ENT, $n = 8$; ENT + αPD-1, $n = 7$; $n = 6$ for PX478 and PX478 + αPD-1. $P$ values were determined by Mantel–Cox test. **d** 4T1 tumor volume in NK-depleted mice ($n = 16$ for control and ENT + αPD-1; PX478 + αPD-1, $n = 10$), T-depleted mice ($n = 16$ for control and ENT + αPD-1; PX478 + αPD-1, $n = 12$) and normal mice (control, $n = 16$; ENT + αPD-1, $n = 14$; PX478 + αPD-1, $n = 10$), at Day 15 of treatments. **e** Lung metastasis of NK-depleted mice (control, $n = 8$; ENT + αPD-1, $n = 7$; PX478 + αPD-1, $n = 5$), T-depleted mice ($n = 7$ for control and ENT + αPD-1; PX478 + αPD-1, $n = 5$) and normal mice ($n = 8$ for control and ENT + αPD-1; PX478 + αPD-1, $n = 6$) at Day 15/30. **f** Flow cytometric analysis of 4T1 tumors. $N = 6$. **g** Flow analysis on HIF1α expression in cells dissociated from 4T1 tumors. $N = 6$. **h, i** Immunofluorescence analysis of 4T1 tumor slides. Representative images (**h** scale bar 20 μm) and quantifications (**i**). $N = 6$. **j, k** Flow cytometric analysis of 4T1 tumors, $n = 5$ for ENT + αPD-1, $n = 6$ for other groups. Data of **f, g, j, k, i** are presented as box and whiskers, with median value and whiskers of minimum and maximum values. Data of **a, b, d, e** were presented as mean ± SD. $P$ values were determined by one-way (**a, b, f, g**) or two-way (**d, e, i, j, k**) ANOVA with Turkey's test. Source data are provided as a source data file.

new therapeutic avenues to overcome the resistance to immunotherapy in TNBC.

The current studies about the role of hypoxia in the tumor immune microenvironment have been controversial. While some studies have shown that hypoxia promotes mouse T cell cytotoxicity[22,23], there are also opposite reports showing that tumor hypoxia induces mouse T cell exhaustion and resistance to immunotherapy and depletion of HIF1α increases NK cell activity and tumor infiltration[16,17,19,48,49]. A recent study has shown that ectopic HIF1α overexpression in mouse T cells diminished the antitumor activity in an adoptive-transfer model[50]. Although the authors did not investigate the mechanism, their data supports our findings that targeting HIF1α in T cells enhanced anticancer immunity. Most importantly, clinical studies have consistently demonstrated that hypoxia is associated with immune exclusion and resistance to immune checkpoint blockade therapy[10,20,21,51]. For example, large-scale clinical studies in melanoma have shown that tumor hypoxia signature is associated with impaired T cell function and resistance to anti-PD-1 therapy[21]. In another study, hypoxia signature gene markers are enriched with non-responders to anti-PD-1 treatment compared to responders[20]. Our in vitro system that allows evaluating the role of hypoxia in human T cell interaction with tumor cells allows mechanistic investigation of hypoxia in regulating the immune response.

Current studies have focused on the role of hypoxia-induced metabolic reprogramming in causing T cell dysfunction and exhaustion[16,19,52–55]. For example, a recent study showed that hypoxia and continued antigen stimulation could induce mitochondrial stress in mouse T cells, leading to terminal exhaustion but in a HIF1α-independent manner[19]. Our in vitro system for studying human immune cell activity shows that hypoxia can induce HIF1α-dependent epigenetic events to directly suppress immune effector expression in both T and NK cells, leading to impaired immune effector dysfunction. In addition to affecting immune effector function, hypoxia, plus continuous tumor antigen stimulation, can also induce expressions of exhaustion markers TIM-3 and TIGIT, which is HIF1α-independent. Thus, it is reasonable to hypothesize that in addition to a direct effect of HIF1α for suppressing immune effectors, hypoxia can also engage HIF1α-independent events to drive T cell exhaustion either through induction of co-inhibitory molecules such as TIM-3 or via other mechanisms such as metabolic stress. Importantly, in both in vitro and in vivo syngeneic and humanized models, we show that targeting the HIF1α-dependent epigenetic mechanism seems sufficient to rescue the T cell dysfunction and sensitizes to PD-1 blockade. In the future, it will be interesting to explore the HIF1α-independent induction of TIM-3 and TIGIT, which may provide an additional strategy to more effectively prevent T cell exhaustion.

Our study uncovered the role of epigenetics in hypoxia-induced immune resistance. We found that hypoxia induces HIF1α and HDAC1 co-occupancy at the effector genes, leading to reductions of H3K27ac and increases of H3K27me3 through a pre-existing PRC2 presence effector gene loci, and suppressed effector gene expression in both T and NK cells. We showed this hypoxia effect on immune cells also leads to inadequate interferon signaling response, including PD-L1 and PD-L2 in TNBC. In both syngeneic and humanized mouse models, PD-1 blockade plus inhibition of HIF1α or HDAC1 is sufficient to increase intratumoral expression of effector genes on CD8+ T cells and NK cells and IFNγ production in tumors that showed a regressive response. Thus, HIF1α-mediated HDAC1/PRC2 provides an epigenetic checkpoint to attenuate effector immune cell function, and targeting this pathway through either genetic depletion or therapeutic intervention may be helpful to overcome immune escape in TNBC where immune checkpoint immunotherapies have limited response.

In this study, the inhibitors we used, PX478, ENT, and EPZ are either in clinical development or clinically approved. Using both syngeneic and humanized mouse models, we demonstrated the efficacy of PD-1 blockade combined with PX478 and ENT to suppress tumor growth and metastasis in anti-PD-1-resistant TNBC, which holds very promising potential in the future treatment of metastatic TNBC. Of note, in a recent report, ENT plus DNA demethylation agent AZA was found to promote the migration of myeloid cells to the metastatic niche to suppress metastasis in different tumor models[56]. Admittedly, there is no doubt that PX478 and epigenetic drugs also have additional effects on other elements such as myeloid cells and macrophages in the tumor microenvironment. Nevertheless, we have validated that the combination treatments significantly promoted CD8+ T cell infiltration in hypoxic tumor areas and increased IFNγ expression in the intratumoral CD8+ T and NK cells. Moreover, depletion of T or NK cells abolished the combination effect, further supporting our claim that the synergistic effect act by removing the hypoxia-mediated epigenetic suppression of immune effector expression.

In summary, our findings add to our fundamental understanding of hypoxia regulation of immune response, which aids in providing alternative strategies for developing a therapeutic intervention to treat TNBC. Furthermore, given that hypoxia is a common trait for solid tumors, we believe that both the mechanism and the combinational therapy we have identified are expandable to the treatment of other different tumor types in addition to TNBC.

## Methods

**Study approval.** For human PBMC samples used in this study, blood donors were recruited in the study approved by the National Health Group (NHG) Domain

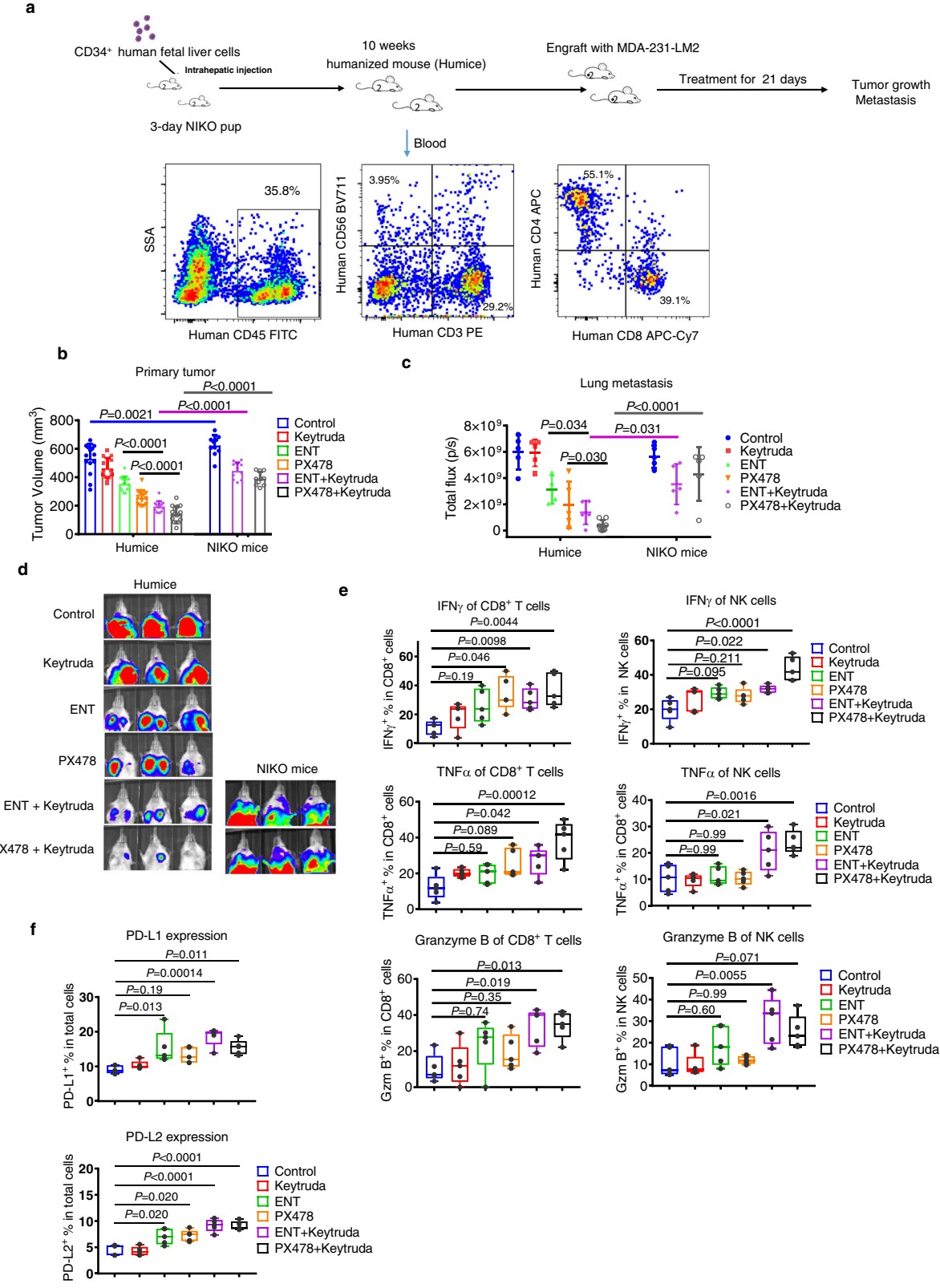

Specific Review Board with reference number 2019/00217-SRF0002. All the studies with these samples were under informed consent and approved for research purposes by the Ethics Committee of Tan Tok Seng Hospital (Singapore). Human TNBC FFPE slides were provided by the Department of Cancer and Inflammation Research, University of Southern Denmark (Denmark) and Saint John's Cancer Institute (CA, USA). Studies with the materials were approved by the Research Ethics Committee of the University of Southern Denmark and the Institutional Review Board of Saint John's Cancer Institute, respectively. Informed written consent was acquired from each individual who agreed to provide tissue samples for research purposes.

**Cell lines and reagents**. The human triple-negative breast cancer cell lines BT549, MDA-MB-231, and mouse triple-negative breast cancer cell line 4T1 were purchased from American Type Culture Collection (ATCC, USA). MDA-MB-231-LM2 was a kind gift from Dr. Yibin Kang, Princeton University, USA. All the cell lines were infected with lentivirus packaged with pLenti-V5-luc plasmid and

**Fig. 7 Pharmacological inhibition of HIF1α and associated epigenetic events sensitizes PD-1 blockade in a humanized mouse model. a** Schematic diagram showing the establishment of humanized mice (humice) with human immune system reconstituted in NIKO mice. The presence of human CD45[+] cells, NK cells, CD4[+] and CD8[+] T cells in the mice's peripheral system was validated by flow cytometry. **b** Primary LM2 tumor size in humice (control, $n = 14$; Keytruda, $n = 14$; ENT, $n = 12$; PX478, $n = 14$; ENT + Keytruda, $n = 16$; PX478 + Keytruda, $n = 16$) and NIKO mice (control, $n = 10$; ENT + Keytruda, $n = 10$; PX478 + Keytruda, $n = 10$), at Day 21 of treatments. **c** Lung metastasis of humice (control, $n = 6$; Keytruda, $n = 6$; ENT, $n = 6$; PX478, $n = 6$; ENT + Keytruda, $n = 7$; PX478 + Keytruda, $n = 7$) and NIKO mice (control, $n = 5$; ENT + Keytruda, $n = 5$; PX478 + Keytruda, $n = 5$) bearing LM2 tumors at Day 35 assessed by bioluminescence (BLI) measurement. **d** Representative bioluminescence (BLI) images showing the lung metastasis of humice and NIKO mice. **e** Flow cytometric analysis of LM2 tumors harvested from humanized mice. IFNγ, TNFα, and granzyme B expression was examined in tumor-infiltrating human CD8[+] T cells and NK cells. $N = 5$ for each group. **f** Flow cytometry analysis of LM2 tumors harvested from humanized mice. Expressions of human PD-L1 and PD-L2 were examined in total living cells dissociated from LM2 tumors. $N = 5$ for each group. Quantification data of flow cytometry (**e**, **f**) are presented as a box and whiskers, with median values and whiskers of minimum and maximum values. Data for **b** and **c** were presented as mean ± SD. $P$ values were determined by one-way (**e**, **f**) or two-way (**b**, **c**) ANOVA with Turkey's test. Source data are provided as a source data file.

selected under blasticidin to generate cells stably expressing luciferase. BT549, MDA-MB-231, and MDA-MB-231-LM2 cells were grown in DMEM (Invitrogen, USA) supplemented with 10% fetal bovine serum (FBS) (Thermo Fisher Scientific, USA). 4T1 cells were maintained in RPMI-1640 (Invitrogen) supplemented with 10% FBS. All media were supplemented with 5000 U/ml penicillin-streptomycin (Invitrogen). NK92-MI cell line was purchased from ATCC and cultured in Alpha MEM (Invitrogen) supplemented with 12.5% FBS, 12.5% horse serum, 1% MEM Vitamin (Invitrogen), and 0.1 mM Mercaptoethanol (Sigma-Aldrich, USA).

Entinostat (CAT# S1053) and PX478 (CAT# S7612) was purchased from Selleck Chemicals, USA. EPZ6438 (CAT# GC14062) was purchased from Glpbio Technology, USA. Digoxin (CAT# 31700) was purchased from Sigma-Aldrich, USA. Mouse PD-1 blockade antibody (CAT# BE014) was purchased from Bio X Cell, USA.

**Expansion and isolation of primary immune cells from human PBMC.** Human peripheral blood mononuclear cells (PBMC) were isolated from the blood using Ficoll Paque Plus (Sigma-Aldrich). The subtypes of MHC I molecules of Human PBMC were examined and HLA-A2[+] human PBMC were identified by flow cytometry. Primary T cells were activated and expanded by culturing $1 \times 10^6$ PBMC in TexMACS medium (Miltenyi Biotec, Germany) supplemented with 500 IU/ml human recombinant Interleukin 2 (rhIL2) (Miltenyi Biotec), with 10 μl TransAct (Miltenyi Biotec) which contains anti-CD3 and CD28 antibodies, for 3 days. The T cells were then purified with a pan-T cell isolation kit (Miltenyi Biotec) and cultured in TexMACS medium + rhIL2 for 2 weeks before CD3/28 antibody re-stimulation or DC cell-based antigen stimulation. Primary NK cells were activated and expanded by culturing $1 \times 10^6$ PBMC in NK MACS medium (Miltenyi Biotec) supplemented with 5% human AB serum (Invitrogen) and 500 IU/ml rhIL2, with $5 \times 10^5$ microbeads conjugated with anti-CD335 and CD2 antibodies (Miltenyi Biotec) for 7 days. The NK cells were purified with NK cell isolation kit (Miltenyi Biotec) and cultured in NK MACS medium + human AB serum + rhIL2 for 1 week before hypoxic culture. The purity of the immune cells was examined by flow cytometry analysis.

**Human DC cell expansion and TCL stimulation.** HLA-A2[+] human PBMC were cultured in RPMI-1640 medium supplemented with 10% FBS, 500 IU/mL rhIL2, 20 ng/mL recombinant human Interleukin 4 (rhIL4) (Miltenyi Biotec), and 20 ng/mL recombinant human granulocyte-macrophage colony-stimulating factor (rhGM-CSF) (Miltenyi Biotec) for 7 days for DC cell expansion. The PBMC with enriched DC cells were primed with 100 ng/ml MHSP65 protein Biovison, USA) and 50 μg/ml TNBC cell lysate (TCL) for further DC expansion and activation. After 3 days of TCL priming, the DC cells were ready to use for T cell stimulation. TCL was prepared through the freeze-and-thaw cycle of TNBC cell suspension in PBS buffer.

**Hypoxic culture of human T cells and NK cells.** For the CD3/28 antibody re-stimulation model, human pan-T cells, which had been expanded for 2 weeks after the first stimulation, were re-stimulated with 5 μL TransAct (CD3/28 antibodies) per $10^6$ cells for 3 days. CD3/28 antibodies were removed (starting point, day 0) and T cells were cultured at $2 \times 10^5$ cells/well in 24-well plate, in 1 mL TexMACS medium + 500 IU/ml rhIL2, under normoxia (20% $O_2$) or hypoxia (1% $O_2$). After 48 h (day 2), the medium was changed with fresh TexMACS medium + rhIL2. After another 48 h (day 4), the T cells were harvested for molecular analysis.

For the DC stimulation model, human HLA-A2[+] pan-T cells which had been expanded for 2 weeks after the first stimulation were cultured at $2 \times 10^5$ cells/well in a 24-well plate together with $2 \times 10^4$ TCL-primed DC cells, in 1 mL TexMACS medium + 500 IU/ml rhIL2 (starting point, day 0), under normoxia (20% $O_2$) or hypoxia (1% $O_2$). After 48 h (day 2), the cells were harvested and $1 \times 10^5$ T cells were seeded onto a monolayer of $2 \times 10^5$ TNBC cells in 2 ml fresh TexMACS medium + rhIL2 in a 12-well plate. After another 48 h (day 4), the T cells in suspension were gently collected from the coculture for flow cytometry or RNA

extraction and the TNBC cells were washed twice with cold PBS and harvested for RNA extraction or western blot.

For the NK cell hypoxic culture, human primary NK cells were cultured $2 \times 10^5$ cells/well in a 24-well plate, in 1 ml NK MACS medium + human AB serum + rhIL2 (starting point, day 0) under normoxia (20% $O_2$) or hypoxia (1% $O_2$). After 48 h (day 2), the NK cells were harvested and $5 \times 10^4$ NK cells were seeded onto a monolayer of $5 \times 10^5$ TNBC cells in 2 ml fresh NK MACS medium, supplemented with 5% human AB serum, 500 IU/ml rhIL2, 10 ng/ml recombinant human Interleukin 12 (rhIL12) (STEMCELL Technologies, USA), 10 ng/ml recombinant human Interleukin 18 (rhIL18) (STEMCELL Technologies), and 5 ng/ml recombinant human Interleukin 15 (rhIL15) IL15 (STEMCELL Technologies), in a six-well plate. After another 48 h (day 4), the NK cells in suspension were gently collected from the coculture for flow cytometry or RNA extraction and the TNBC cells were washed twice with cold PBS and harvested for RNA extraction or western blot.

Preliminary experiments were performed to optimize the drug concentrations. For the drug treatments on T and NK cells, PX478, Digoxin, Entinostat, or EPZ6438 were added into the cell suspension at 10 μM, 50 nM, 200 nM, and 500 nM, respectively at day 0. After 48 h (day 2), drugs were replenished at the same concentration when the T or NK cells were resuspended in a fresh medium. Drug treatments at the indicated doses did not show toxicity to the T or NK cells.

The processing of hypoxic cultured immune cells was performed in a hypoxia chamber at 1% oxygen. Human T or NK cells were resuspended, centrifuged, washed and lysed for RNA extraction, western blot, or immunoprecipitation, or fixed in 1% paraformaldehyde for Chromatin-IP inside the hypoxia chamber. For flow cytometry, human T or NK cells were collected in an ice-cold 1.5 ml tube and incubated on ice for 5 min before the cells were transferred from the hypoxia chamber to the bench.

**RNA-seq.** Human pan-T or NK cells were collected from coculture and lysed directly with Trizol (Invitrogen) as described above. Total RNA was isolated and purified with the RNeasy Mini Kit (Qiagen, German). The RNA samples were sent to Novogene AIT (Singapore) for sequencing and mRNA library generation.

For data processing and analysis, raw fastq files were examined for quality using FastQC (v0.11.9) and MultiQC (v1.10), then the paired fastq files were aligned to human genome hg38 using HISAT2 (v2.2.1). After alignment, the BAM files were inspected for various alignment metrics using RSeQC and gene-level counts were quantified from BAM files using featureCounts (v2.0.2). The raw count data were then normalized and rescaling to counts per 1 million (CPM). Raw count data were subsequently imported into R (v4.0.3) and edgeR (v3.32.1) package were used to identify the differentially expressed gene. Thresholds for differential expression were |LogFC| > 1 and adj. $P$ value < 0.01. For heatmap generation, the R package ComplexHeatmap (v2.6.2) were used and the two annotation gene list "POSITIVE_REGULAATION_OF_CYTOKINE_PRODUCTION" and "CYTOKINS_AND_GROWTH FACTORS" were downloaded from MsigDB database. For gene set enrichment analysis (GSEA), gene signatures were obtained from the MSigDB "Hallmark gene sets" and "Gene Ontology gene sets" collections. All expressed genes were pre-ranked by logFC value (Hypoxia/Normal, performed with edgeR as previously described). GSEA analysis was then performed using the clusterProfiler (v3.18.1) package. Raw RNA-seq data were uploaded in GEO with accession code GSE179885.

**Mouse T cell expansion and cultures.** All mouse experiments were conducted in compliance with animal protocols approved by the ASTAR-Biopolis Institutional Animal Care and Use Committee of Singapore. The BALB/c mice bearing 4T1 tumors were sacrificed when the tumor reached 500 mm$^3$. The spleens were harvested and mechanically disrupted. The cell suspensions were filtered through a 70-um constrainer to obtain single cells. Splenocytes were isolated from the spleen cell mixture using Ficoll Paque Plus. Mouse T cells were isolated from splenocytes using pan-T cell isolation kit II (Miltenyi Biotec) and stimulated with mouse CD3/CD28 dynabeads in the T cell Activation/Expansion kit (Miltenyi Biotec). After 5 days of stimulation, the CD3/CD28 dynabeads were removed magnetically and

the mouse pan-T cells were expanded in TexMACS medium supplemented with 10% FBS and 500 IU/ml rhIL2 for 2 weeks.

For the hypoxic culture, mouse pan-T cells were cultured in 24-well plates at $1 \times 10^5$ cells/well in 1 ml TexMACS medium + rhIL2 (starting point, day 0) under normoxia (20% $O_2$) or hypoxia (1% $O_2$). After 48 h (day 2), the medium was changed with fresh TexMACS medium + rhIL2. After another 48 h (day 4), the cells were harvested and $1 \times 10^5$ T cells were seeded onto a monolayer of $2 \times 10^5$ mouse 4T1 cells in 2 ml fresh TexMACS medium + rhIL2 in a 12-well plate. After another 48 h (day 4), the mouse T cells in suspension were gently collected from the coculture for flow cytometry.

For the drug treatments on mouse T cells, PX478, Digoxin, Entinostat, or EPZ6438 were added into the cell suspension at 20 μM, 100 nM, 500 nM, and 1 μM respectively at day 0. After 48 h (day 2), drugs were replenished at the same concentration when mouse T cells were resuspended in a fresh medium. Preliminary experiments were performed to optimize the drug concentrations. Drug treatments at the indicated doses did not show toxicity to mouse T cells.

**Cell lysis assay**. Human NK or antigen-stimulated T cells were cultured at $2 \times 10^5$ cells/well in 24-well plate under normoxia or hypoxia for 72 h before coculture with a TNBC monolayer. Human TNBC cell lines stably expressing luciferase were seeded in 96-well plates with a white bottom at 1000 cells/well and cultured under normoxia for 24 h. Medium of TNBC cells in 96-well plates was removed and human NK or antigen-stimulated T cells were resuspended in DMEM + 10% FBS and seeded onto the TNBC monolayer at indicated cell number. The cells were cocultured under normoxia or hypoxia for another 48 h. The luciferase substrate luciferin (Promega, USA) was added into the wells at the end of the experiment and luminescent signals were detected in GloMAX Explorer (Promega).

**CRISPR-based gene knockdown in T cells**. Transfection of human T cells with Caspase 9-sgRNA (RNP) complex was conducted with the 4D-Nucleofector X kit (Lonza, Switzerland) according to the manufacturer'smanufacturer's instructions. Briefly, the RNP complex was generated by incubating Caspase 9 protein with sgRNA at a molar ratio of 1:3 for 20 min under room temperature. RNP complex was then mixed with human T cells at 150 pmol sgRNA per million cells in the Nucleofector Solution (Lonza). Electroporation was done with the 4D-Nucleofector X Unit (Lonza) under the program E0-115. Cells were suspended and cultured in TexMACS medium + 500 IU/ml rhIL2 after electroporation. Sequences of sgRNA were summarized in Supplementary Table 3.

**Immunofluorescence**. Human TNBC FFPE slides were provided from the Department of Cancer and Inflammation Research, University of Southern Denmark (Denmark), and Saint John's Cancer Institute (CA, USA). Mouse 4T1 tumors were harvested and fixed immediately in 10% formaldehyde for 48 h and then washed with 75% ethanol. Mouse tumors were embedded in paraffin and sectioned to a glass slide. The human TNBC slides and mouse 4T1 tumor slides were deparaffinized, rehydrated, and antigens were retrieved by pH 9.0 Tris-EDTA buffer. Slides were blocked with 5% normal goat serum in PBS for 20 min. Human TNBC samples were incubated with rabbit anti-human HIF1α antibody (Abcam, CAT# ab51608, dilution 1:100), mouse anti-human CD8 antibody (Cell Signaling Technology, CAT#70309, dilution 1:100), and rat anti-human/mouse IFNγ antibody (Invitrogen, CAT# MM700, dilution 1:50) overnight under 4 °C. The slides were washed with 0.1% Tween20 in PBS three times and then incubated with secondary antibodies at 1:100, including anti-rabbit Alex Fluor 647 (Invitrogen, CAT# A32733), anti-mouse Alex Fluor 488 (Invitrogen CAT# A32723) and anti-rat Alex Fluor 546 (Invitrogen, CAT# A11081). Mouse 4T1 tumor samples were incubated with FITC-conjugated mouse anti-activated pimonidazole antibody (Hypoxyprobe, CAT# HP2-200kit, dilution 1:100), rabbit anti-mouse CD8a antibodies (Cell Signaling Technology, CAT# 98914, dilution 1:100), rat anti-human/mouse IFNγ antibody (Invitrogen, CAT# MM700, dilution 1:50) overnight under 4 °C. The slides were washed with 0.1% Tween20 in PBS three times and then incubated with secondary antibodies at 1:100, including anti-rabbit Alex Fluor 647 (Invitrogen, CAT# A32733) and anti-rat Alex Fluor 594 (Invitrogen, CAT# A11007). After washing with PBS, both human TNBC and mouse tumor slides were stained with DAPI and mounted in FluorSave (Millipore, CAT# 345789). For each sample, quantifications were based on four images taken at random fields using a Leica Lightsheet microscope. Fluorescent signals were captured at the wavelength of 660–700 nm for Alex Fluor 647, 500–540 nm for Alex Fluor 488, 580–620 nm for Alex Fluor 546, and 600–630 nm for Alex Fluor 594. Images were analyzed in the software of Leica Application Suite X.

**Quantitative RT-PCR**. Total RNA was isolated by using TRIzol (Invitrogen) and purified with the RNeasy Mini Kit (Qiagen). Reverse-transcription and quantitative PCR assays were performed using the High Capacity cDNA Archive kit and KAPA SYBR Fast qPCR kit (KAPA Biosystems, USA). For quantification of mRNA levels, ACTB and GAPDH level was used as an internal control. Primers for qPCR were summarized in Supplementary Table 1. All reactions were analyzed in an Applied Biosystems ViiA7 PCR system using the software QuantStudio v1.7.1 in 96-well or 384-well plate format.

**TCGA data analysis**. The dataset "BRCA.exp.547.med.txt" was obtained from the TCGA breast cancer online portal (https://www.cancer.gov/about-nci/organization/ccg/research/structural-genomics/tcga). Breast cancer subtypes were classified based on the PAM50 gene signatures. Basal (TNBC), $n = 98$; HER2+, $n = 58$; Luminal A, $n = 231$; Luminal B, $n = 129$. Data were processed and Pearson's correlation coefficients were calculated in R (v4.0.3). Correlation heatmap and scatter plots were generated in R (v4.0.3).

**Immunoblotting**. All immunoblotting analyses were performed following SDS PAGE using standard methods. Antibodies used for immunoblotting were HIF1α antibody (Abcam, CAT# ab51608, dilution 1:1000), HIF1α antibody (Cell Signaling Technology, CAT# 79233, dilution 1:1000), EZH2 antibody (Cell Signaling Technology, CAT# 5246, dilution 1:1000), SUZ12 antibody (Cell Signaling Technology, CAT# 3737, dilution 1:1000), EED antibody (Millipore, CAT# 09-744, dilution 1:1000), HDAC1 antibody (Cell Signaling Technology, CAT# 34589, dilution 1:1000), HDAC2 antibody (Cell Signaling Technology, CAT# 57156, dilution 1:1000), phosphorylated-STAT1 antibody (Cell Signaling Technology, CAT# 9167, dilution 1:1000), STAT1 antibody (Cell Signaling Technology, CAT# 9176, dilution 1:1000), PD-L1 antibody (Cell Signaling Technology, CAT# 3684, dilution 1:1000), IRF1 antibody (Abcam, CAT# ab26109, dilution 1:1000), H3K27me3 antibody (Cell Signaling Technology, CAT# 9733, dilution 1:1000), H3K27ac antibody (Abcam, CAT# 4729, dilution 1:1000), and GAPDH antibody (Cell Signaling Technology, CAT# 2118, dilution 1:1000).

**Co-Immunoprecipitation**. The procedure of coco-immunoprecipitation assays as described previously[57]. Briefly, cells were lysed using IP buffer (20 mM Tris pH 7.5, 100 mM NaCl, 0.5% NP-40, 0.5 mM EDTA) supplemented with PMSF, NaF, Na3VO4, and proteinase inhibitors. Cell lysates were precleared with Protein G/A agarose bead (Santa Cruz) for 4 h and incubated with HIF1α antibody (Abcam, CAT# ab51608, dilution 1:100), SUZ12 antibody (Active Motif, CAT# 39357, dilution 1:100) and normal rabbit IgG (Santa Cruz, CAT# sc2027, dilution 1:100). The precipitated protein complex was captured using protein A-Agarose beads (Roche, USA) and extensively washed with the washing buffer (50 mM Tris-HCl, 150 mM NaCl, 0.1% Triton). The precipitated proteins were dissolved in SDS sample buffer along with 3 mM dithiothreitol and subjected to immunoblotting analysis.

**Chromatin immunoprecipitation (ChIP)**. The protocol of ChIP assay was described previously[58]. Briefly, immune cells were cross-linked using 1% formaldehyde for 10 min. The reactions were stopped by adding 0.125 mM glycine and the cells were washed three times using ice-cold TBSE buffer (20 mM Tris-HCl, pH 7.5, 150 mM NaCl, and 1 mM EDTA). Cells were then lysed in SDS lysis buffer (50 mM Tris-HCl, pH 8.0, 10 mM EDTA, and 1% SDS) and sonicated (25% amplitude, 10 s on and 20 s off for 16 cycles) to achieve a DNA shear length of 200–500 bp. Solubilized chromatin was diluted ten times in dilution buffer (10 mM Tris-HCl, pH 7.4, 140 mM NaCl, 1 mM EDTA, 1% Triton X-100, and 0.01% SDS) and precleared with Protein G agarose beads (Santa Cruz, USA) for 4 h at 4 °C. The following antibodies were used for immunoprecipitation with the chromatin: H3K27 acetylation (Abcam, CAT# ab4729; dilution 1:100), H3K27me3 (Cell Signaling Technology, Cat# 9733, dilution), HIF1α (Abcam, CAT# ab51608; dilution 1:100), HDAC1 (Abcam, CAT# ab7028, dilution 1:100), HIF2 α (Abcam, CAT# ab199, dilution 1:100), HDAC3 (Cell Signaling Technology, CAT# 85057, dilution 1:100), SUZ12 antibody (Active Motif, CAT# 39357, dilution 1:100), EZH2 antibody (Cell Signaling Technology, CAT# 5246, dilution 1:100) and HDAC2 (Abcam, CAT# ab7029, dilution 1:100). The complexes of were captured with Dynabead (Life Technologies,USA) at 4 °C for 2 h and washed three times with dilution buffer and once with TE buffer (10 mM Tris-HCl, pH 7.5and 1 mM EDTA), sequentially. Elution and reverse cross-linking were performed in elution buffer (50 mM Tris-HCl, pH 7.5, 10 mM EDTA, and 1% SDS) at 68 °C with vortex overnight followed by digestion with proteinase K (20 mg/ml) at 42 °C for 2 h. Eluted DNA was purified using the E.Z.N.A. Cycle Pure Kit (Omega). Precipitated chromatin DNA was analyzed by real-time quantitative PCR. ChIP-qPCR primers of RPL30 were purchased from Cell Signaling Technology (CAT# 7014). Other primers for ChIP-qPCR were summarized in Supplementary Table 2.

**Flow cytometry**. For flow cytometry analysis of in vitro human immune cells, the human T or NK cells were incubated with 1× Golgiplug (BD Biosciences, USA) for 6 h. Suspensions containing immune cells were stained with surface antibodies in FACS buffer (PBS with 0.5% BSA and 0.1% sodium azide) for 30 min. Fixation and permeabilization process was carried out with fixation buffer (BD Biosciences) according to the manufacturer's protocols. The immune cells were then stained with antibodies in washing buffer (BD Biosciences) to detect intracellular proteins. The following antibodies were used for staining, anti-human CD3 Vioblue antibody (Miltenyi Biotec, CAT# 130-114-519, dilution 1:100), anti-human CD3 FITC antibody (BD Biosciences, CAT# 555339, dilution 1:100) anti-human CD8 APC-Cy7 antibody (BD Biosciences, CAT# 557834, dilution 1:100), anti-human CD8 FITC antibody (Biolegend, CAT# 334704, dilution 1:100), anti-human CD4 PE antibody (BD Biosciences, CAT# 555347, dilution 1:100), anti-human TIM-3 APC antibody (Miltenyi Biotec, CAT# 130-120-700, dilution 1:100), anti-human LAG-3

Vioblue antibody (Miltenyi Biotec, CAT# 130-118-549, dilution 1:100), anti-human TIGIT PE-Vio770 (Miltenyi Biotec, CAT# 130-116-817, dilution 1:100), anti-human PD-1 PE-Vio770 (Miltenyi Biotec, CAT# 130-120-391, dilution 1:100), anti-human CTLA-4 FITC antibody (Miltenyi Biotec, CAT# 130-116-809, dilution 1:100), anti-human TOX APC antibody (Miltenyi Biotec, CAT# 130-118-335, dilution 1:100), anti-human IFNγ BV785 (Biolegend, CAT# 502542, dilution 1:100), anti-human TNFα BV650 (BD Biosciences, CAT# 563418, dilution 1:100), anti-human granzyme B BV421 (BD Biosciences, Cat# 563389, dilution 1:100), anti-human IFNγ PE-Vio770 (Miltenyi Biotec, CAT# 130-113-494, dilution 1:100), anti-human TNFα APC (Miltenyi Biotec, CAT# 130-120-063, dilution 1:100), anti-human TIM-3 BV650 antibody (Biolegend, CAT# 345028, dilution 1:100), anti-human PD-1 BV711 antibody (Biolegend, CAT# 329928, dilution 1:100).

For cells dissociated from mouse 4T1 tumors or human LM2 tumors, cells were incubated with 10 nM PMA (Sigma-Aldrich) and 1 μM ionomycin (Sigma-Aldrich) in PBS under room temperature for 20 min. The mouse and human cells were incubated with 1× Live/Dead Fixable Dead Cell dye (Invitrogen) in ice-cold PBS for 20 min and blocked in anti-mouse FcR antibody (Miltenyi Biotec) and anti-human FcR antibody (Miltenyi Biotec) respectively in ice-cold FACS buffer for 20 min. The cells were then processed for antibody staining as describe above. The following antibodies were used for staining, anti-activated pimonidazole FITC antibody (Hypoxyprobe, CAT# HP2-200kit, dilution 1:200), anti-mouse HIF1α APC antibody (R&D Systems, CAT# IC1935A, dilution 1:50), anti-mouse CD3 BV421 antibody (BD Biosciences, CAT# 564008, dilution 1:100), anti-mouse CD45 Percp-Vio700 antibody (Miltenyi Biotec, CAT# 130-110-663, dilution 1:100) anti-mouse CD8 APC-Vio770 antibody (Miltenyi Biotec, CAT# 130-120-737, dilution 1:100), anti-mouse Nkp46 APC antibody (Miltenyi Biotec, CAT# 130-112-202, dilution 1:100), anti-mouse CD4 BV650 antibody (Biolegend, CAT# 563747, dilution 1:100), anti-mouse TIM-3 BV711 antibody (Biolegend, CAT# 119727, dilution 1:100), anti-mouse PD-1 PE-Vio770 (Miltenyi Biotec, CAT# 130-120-391, dilution 1:100), anti-mouse IFNγ PE (Miltenyi Biotec, CAT# 130-117-352, dilution 1:100), anti-mouse TNFα BV711 (BD Biosciences, CAT# 563944, dilution 1:100), anti-mouse/human granzyme B FITC (Miltenyi Biotec, CAT#130-118-430, dilution 1:100), anti-mouse PD-L1 BV786 antibody (BD Biosciences, CAT# 741014, dilution 1:100), anti-mouse PD-L2 FITC antibody (Miltenyi Biotec, Cat# 130-102-222, dilution 1:100), anti-human CD45 FITC antibody (BD Biosciences, CAT# 304006, dilution 1:100), anti-human CD3 PE antibody (Biolegend, CAT# 300308, dilution 1:100) anti-human CD8 APC-Cy7 antibody (BD Biosciences, CAT# 557834, dilution 1:100), anti-human CD56 BV711 antibody (Biolegend, CAT# 318336, dilution 1:100), anti-human CD4 APC antibody (Biolegend, CAT# 300514, dilution 1:100), anti-human IFNγ BV785 (Biolegend, CAT# 502542, dilution 1:100), anti-human TNFα BV650 (Biolegend, CAT# 502398, dilution 1:100), anti-human Granzyme B BV421 (BD Biosciences, Cat# 563389, dilution 1:100), anti-human PD-L1 PE-Cy7 antibody (Biolegend, CAT# 374506, dilution 1:100), anti-human PD-L2 PE antibody (Miltenyi Biotec, CAT# 130-098-530, dilution 1:100).

The flow cytometry was run using LSRFortessa- X20 flow cytometer (BD Biosciences) or MACSQuanty X flow cytometer with the software BD FACSDiva 6.0 and the results were analyzed with Flow Jo v10.7.1.

### Human CD34+ cells and NIKO, immune, and TNBC-immune-humanized mice.

Human umbilical cord blood was collected from KK Women's and Children's Hospital (KKH), with written consent obtained from guardians of donors, and in accordance with the ethical guidelines of KKH. Human CD34+ cells were isolated and purified as described earlier.

All manipulations and procedures with mice were approved by Agency for Science, Technology and Research (A*STAR) Institutional Animal Care and Use Committee (IACUC). All mice were maintained at 21 °C ± 1, 55 to 70% humidity, and with a 12 h light/ dark cycle, from 7 a.m to 7 p.m. Mice were housed in a sterile environment and only accessed under a BSL2 hood. Mice were fed, given water, and monitored daily for health, and cages were changed weekly. NIKO mice (Immunodeficiency, NOD/SCID/IL2Rg−/−) were generated and bred by Biological Resource Centre (BRC), A*STAR, Singapore. For immune-humanized mice, one to three days old NIKO pups were irradiated with a 55 s exposure equaling 1.1 Gy and transplanted with $1 \times 10^5$ CD34+ human cord blood cells (HLA-A2+) by intra-hepatic injections[46]. The mice were bled at 12 weeks post-transplantation to determine the fraction of human immune cell reconstitution. Immune-humanized mice reconstituted with 20–50% of human CD45+ cells were used for engraftment[59]. To generate TNBC-immune-humanized mice, fresh MDA-231-LM2 cells (HLA-A2+) labeled with luciferase were harvested and resuspended in 1:1 of matrigel (BD Bioscience) and engrafted ($2 \times 10^6$ cells in 50 ul of mix) into appropriate female immune-humanized mice by orthotopic injection (mammary fat pad).

For the drug administration, mice were randomized and drug treatments started 15 days after tumor injection when the tumor volume reached 70–100 mm³. Entinostat (ENT) was given by intraperitoneal injection at the dose of 10 mg/kg, daily. PX478 was given by intraperitoneal injection at the dose of 40 mg/kg, daily. Keytruda (pembrolizumab) was given by intraperitoneal injection at 10 mg/kg for the first dose, 5 mg/kg for the following doses, twice a week. Entinostat was dissolved in the formulation of 4% DMSO, 40% PEG300, 5% Tween-80, and 51% PBS. PX478 was dissolved in the formulation of 4% DMSO and 96% PBS. Keytruda was diluted in PBS. Tumor volume was measured twice a week since the beginning of drug treatments. Tumor volume was calculated as $V = L \times W \times W/2$. To monitor metastasis, mice were injected with luciferin intraperitoneally and luminescence were measured with IVIS Spectrum In Vivo Imaging System (PerkinElmer, USA). To access metastasis to lung, the luminescence at the mouse chest was examined and quantified. The maximal tumor size, 1000 mm³, was not exceeded as requested by the Institutional Animal Care and Use Committee of Singapore.

For ex vivo analysis, LM2 tumors of humanized mice were harvested by surgery with proper anesthesia, on Day 21 of drug treatment. Tumors were dissociated with Human Tumor Dissociation Kit (Miltenyi Biotec) following the manufacturer's instructions and filtered with a 70-μm strainer to get single cells. The dissociated cells were then processed for flow cytometry analysis.

### Syngeneic mouse TNBC model.

All experiments were conducted in compliance with animal protocols approved by the ASTAR-Biopolis Institutional Animal Care and Use Committee of Singapore. Six- to eight-week-old female BALB/c mice were purchased from InVivos (Singapore). All mice were maintained at 21 °C ± 1, 55 to 70% humidity, and with a 12 h light/ dark cycle, from 7 a.m. to 7 p.m. To establish an orthotopic mouse TNBC model, 4T1-Luc cells were harvested and resuspended in 1:1 matrigel (BD Bioscience) and injected ($5 \times 10^4$ cells in 50 ul of mix) into the fat pad of mice.

For the drug administration, mice were randomized and drug treatments started 10 days after tumor injection when the tumor volume reached 70–100 mm³. EPZ6438 was given by intraperitoneal injection at the dose of 50 mg/kg, daily. Entinostat (ENT) was given by intraperitoneal injection at the dose of 10 mg/kg, daily. PX478 was given by intraperitoneal injection at the dose of 40 mg/kg, daily. Mouse PD-1 blockade antibody (BioXcell, CAT# BE014) was given by intraperitoneal injection, 10 mg/kg, every other day. Entinostat and EPZ6438 were dissolved in the formulation of 4% DMSO, 40% PEG300, 5% Tween-80, and 51% PBS. PX478 was dissolved in the formulation of 4% DMSO and 96% PBS. Mouse PD-1 blockade antibody was diluted in PBS. The tumor size was measured twice a week. Tumor volume was calculated as $V = L \times W \times W/2$. Lung metastasis was examined by bioluminescence measurement in IVIS Spectrum In Vivo Imaging System as described above. The maximal tumor size, 1000 mm³, was not exceeded as requested by the Institutional Animal Care and Use Committee of Singapore.

For the depletion of NK and T cells in Balb/C mice, anti-asialo GM1 NK depleting antibody (Thermo Scientific), and/or anti-CD4/CD8 T cell depleting antibody (STEMCELL Technologies) was given by intraperitoneal injection 1 day before the tumor inoculation and 7 days after tumor inoculation. The depletion of NK or T cells were examined by flow cytometry of PBMC extracted from mouse blood.

For ex vivo analysis, BALB/c mice were sacrificed and 4T1 tumors were harvested on Day 21 of drug treatment. Tumors were either directly fixed in 10% formaldehyde or dissociated with Mouse Tumor Dissociation Kit (Miltenyi Biotec) following the manufacturer's instructions and filtered with a 70-μm strainer to get single cells. The fixed tumor samples were processed for immunofluorescence and the dissociated cells were processed for flow cytometry analysis.

### Statistics.

All statistical analyses were performed in GraphPad Prism version 9.0. For multiple comparisons, the $P$ values were determined with one-way analysis of variance (ANOVA) or two-way ANOVA, as indicated in the figure legends. For comparisons of two groups, the $P$ values were determined with paired or unpaired two-tailed t-test, as indicated in the figure legends. Survival analysis was based on a log-rank test. Values of $P < 0.05$ were considered significant.

**Reporting summary.** Further information on research design is available in the Nature Research Reporting Summary linked to this article.

## Data availability

Raw RNA-seq data is available in the GEO database with accession number GSE179885. The publicly available data used in Figs. 1a and 1b are available in the TCGA database under accession code BRCA.exp.547.med.txt [https://gdc.cancer.gov/about-data/publications/brca_2012]. The publicly available data used in Fig. 1h and Supplementary Figure 1 are available in the KM-Plotter-Breast Cancer [https://kmplot.com/analysis/index.php?p=service&cancer=breast]. The remaining data are available within the Article, Supplementary Information, or Source Data file. Source data are provided with this paper.

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

## Acknowledgements

This work was supported by the National Medical Research Council of Singapore, Open Fund-Individual Research Grant (OFIRG20nov-0085) to Q.Y., and the Agency for Science, Technology and Research of Singapore under its Career Development Award (202D800029) to S.M. The humanized mouse work was supported by National Research Foundation Singapore Fellowship (NRF-NRFF2017-03) to Q.C. We thank Mei Yee Aau, Kakaly Gosh, Moreno Zanardi, Jing Wen Tan, Denise Hui Lun Pheh, Joey Zu'Er Wong,

and Joey Yong Yi Tham for their assistance in the experiments. We thank Dr. Wei Leong Chew for providing the 4D-Nucleofector electroporation system.

## Author contributions

Q.Y. supervised the project and contributed to the design and interpretation of all experiments. S.M. contributed to the design, conduct, and interpretation of all experiments. Q.C., Y.Z., and M.L. provided humanized mouse model and technical advice on xenograft tumor sample preparation and FACS panel design. L.-T.O., Z.J., P.L.L., and J.Y.G. contributed to assay setup and optimization and technical assistance, including FACS, CRISPR gene knockdown, IF and cell culture, PCR, and Western blot analysis. G.O., A.R., Z.N., and W.W. contributed to bioinformatics data analysis. W.C.L. contributed to in vivo mouse experiments. E.Y.T., S.E., H.J.D., M.A.B., and D.S.B.H. provided human PBMCs and breast tumor samples. S.M. and Q.Y. wrote the manuscript with inputs from all co-authors.

## Competing interests

The authors declare no competing interests.
