## [Peer Review File · Nature Communications]

Hypoxia induces HIF1 α -dependent epigenetic vulnerability in triple negative breast cancer to confer immune effector dysfunction and resistance to anti-PD-1 immunotherapyREVIEWER COMMENTS

Reviewer #1 (Remarks to the Author): with expertise in hypoxia/HIF, epigenetics, immunotherapy, cancer

Ma S et al. reported that HIF1a interacted with HDAC1 to control HDAC1 and EZH2 binding to IFNG, TNF, and GZMB genes to repress their gene expression in T cells and NK cells, leading to immune evasion and resistant to immunotherapy. Combined treatment of HDAC1 inhibitor or HIF-1 inhibitor with anti-PD1 enhanced immunotherapy in syngeneic and humanized mouse models of TNBC. Although the story is interesting, the key conclusions are not well supported by the existing data in the manuscript. Some of data are not convincing and their scientific rigor is low. Overall, this manuscript does not meet with NC standards.

Major points:

1. Fig. 1. Hypoxia is known to be uncorrelated with tumor subtypes, grades, and stages. Luminal breast tumors have less immune cell infiltration as compared with TNBC and HER2+ breast tumors. So it is not surprising that a hypoxia gene signature did not correlate with immune cell signatures in luminal breast tumors. These gene correlation data in luminal breast tumors do not support their conclusion in the manuscript that hypoxia is associated with immune escape in breast cancer.
2. Fig. 1h, S1. It is unclear how the authors defined "high" and "low" in the KM analysis. Given the different numbers in both groups, the authors did not use median to separate two groups, which seems biased.
3. Fig. 3e, S3d-f. Hypoxia had no effect on EZH2/SUZ12 enrichment on IFNG and TNF genes, which cannot explain increased H3K27me3 occupancy on these genes. Reduction in H3K27ac is not sufficient to cause H3K27me3 increase as H3K27me3 is dynamically controlled by methyltransferase EZH2 and demethylases (KDM6A/B). Thus, the mechanism of H3K27ac-H3K27me3 interplay under hypoxia was not clearly addressed. In addition, it is unclear how strong EZH2/SUZ12 enrichment is as control IgG is missing in these experiments.
4. Fig.4a. There is no HRE at the IFNG promoter (P3 and P4). It is not convincing that HIF-1a can bind there.
5. Fig. 4e. Given that HIF-1a did not bind to EZH2 and SUZ12, how does HIF-1a KD reduce H3K27me3 enrichment?
6. Fig. 5a. Tumor cells grow slowly on the petri dishes under hypoxia. Is reduced luciferase activity under hypoxia due to slow tumor cell growth?
7. Fig. 7. The authors implanted human LM2 cells after establishment of humanized mouse model. It is surprising that these tumor cells grew similarly to those in immunodeficient mice and were not rejected in this humanized model?
8. Fig. 7e. Combined treatment failed to synergize IFN γ production, which does not support their finding in tumor growth (Fig. 7b).

Minor points:

1. GAPDH was used as a loading control in western blot data, which is not appropriate. GAPDH is a HIF downstream target gene.
2. Fig. 4b. HIF-1a blot (input) is not convincing.
3. Fig. 6h. The data need to be quantified. The hypoxia area is not correctly circled in the CD8 staining image in PX478+aPD-1 group.
4. A spelling error for PD-L1 on page 11, 2nd paragraph.
5. The authors should add page numbers.

Reviewer #2 (Remarks to the Author): with expertise in hypoxia, cancer

While the understanding of complex tumor microenvironment and response to immunotherapy is required to promote therapeutic efficacy, this manuscript determines how hypoxia plays a role in

immune evasion by focusing on triple negative breast cancer (TNBC). Using both in vitro and in vivo systems, authors demonstrate that the interaction between HIF and HDAC suppresses immune effector genes through epigenetic alterations. By targeting HIF and HDAC, authors suggest combination treatments with PD1 therapy, which can be possibly translated into the clinic. Overall this manuscript clearly presents their supporting data with extensive studies. However, there are minor points that need to be addressed to better present their findings.

1. Figure 1b) It is interesting that HER2 overexpressing breast cancer acts in similar way with TNBC. It might need to be mentioned what causes the differences between TNBC, HER2 and Luminal cancer types. Also, authors need to address rationale why they focus more on TNBC but not HER2 overexpressing breast cancer in terms of hypoxia-mediated T cell exhaustion unless they provide data using HER2 positive cells.
2. Figure 1c, 1f,6h) Throughout the paper, authors did not provide any quantitative data of tissue staining. It is not clear how representative staining images are. Quantification needs to be included by comparing intensity or areas of positive staining.
3. Page 6. Indicated figures are mislabelled. Authors indicate Fig S2b and 2d instead of Fig S2c in this sentence. "Hypoxia did not seem to affect the CD4+ and CD8+ T cell compositions, nor did it affect the overall T cell differentiation status, though it slightly decreased the Tem population (Fig. S2c)"
4. Figure 2d. Authors need to indicate what the gene list in type 2 means.
5. Figure 4a. Authors need to include positive control data for their HIF-1 and HIF-2 ChIP assays, since it is not clear whether insignificant changes in HIF-2 ChIP are due to a technical problem of poor antibody binding or an actual HIF-2-promoter binding.
6. Figure 4g. The quality of Western blot needs to be improved since it is hard to recognize the inhibition effects from HIF1 inhibitors.
7. Figure 6a, 6d, 7b, S9a. Authors present tumor growth data inconsistently. Tumor growth curve needs to be included in each figure to compare the effect of the drug combination.
8. Figure 6b, 6e, S9b. It will be more comprehensive if authors include IHC data from lung showing how metastatic lung nodules formed in different ways depending on each treatment. Bioluminescence imaging is a good tool to track tumor growth non-invasively but also, background effect may need to be considered.
9. Figure 7b, 7c. To emphasize the effect of immune system, authors need to include treatment of PX478 and ENT alone to compare with combination treatment, since ENT and PX478 have anti-tumor effects on their own.
10. Although authors noted in the discussion section, it is not clear how hypoxia regulates TIM-3 and TIGIT and what roles they play in the context of this study. Unless authors provide further data suggesting HIF-independent mechanisms altering these factors, they are better to be excluded in this manuscript.
11. The conclusion clearly supports that combination treatment of HDAC or HIF inhibitors with PD1 therapy inhibits primary tumor growth as well as metastasis. It will be more supportive if authors include patient data showing the correlation of hypoxia vs. metastasis or immune effectors vs. metastasis in TNBC patients.

Reviewer #3 (Remarks to the Author): with expertise in breast cancer, cancer immunology/immunotherapy

Hypoxia is common in several solid tumors and contributes to resistance to several therapies, including immunotherapy. Better understanding of the mechanisms governing hypoxia in the tumor microenvironment will allow identifying new targets in order to develop more precise therapeutic approaches.

In this work, Ma et al. highlight the importance of hypoxia in conferring resistance to immunotherapy in triple-negative breast cancer (TNBC). They demonstrate that under a hypoxic tumor microenvironment, tumor cells of a triple negative breast cancer model acquire resistance to immune checkpoint blockade. This process is controlled by an epigenetic mechanism that involves HIF1alpha

but also by a parallel HIF1 α -independent mechanism that leads to the suppression of immune effector gene expression. Using in vitro assays, the authors show that hypoxia and the epigenetics events associated to HIF1 α impair the immune cells cytotoxicity, the interferon signaling, and induces resistance to immunotherapy (such as promoting a resistance to anti-PD1 blockade). In vivo experiments show that targeting HIF1 α and the associated epigenetic machinery overcomes resistance to PD1 blockade, in both syngeneic and humanized models.

Although most results in this manuscript are consistent, and they employ a wide range of experimental settings, some conclusions are not sustained by the experimental data, and the number of independent experiments and biological replicates need to be increased in several settings.

Major issues:

1. It is unclear to me why the authors have selected breast cancer to study the contribution of hypoxia to immunotherapy response. Despite immunotherapy has been recently approved in TNBC, it will be more relevant to perform their studies in melanoma or lung cancer, where immunotherapy has become the standard of care. Analyses and experiments in these tumor types will be more relevant to the clinical practice. The title of the manuscript does not mention "breast cancer" and in the abstract the authors extrapolate the results to other solid tumors, but only one mouse and two human breast cancer cell lines were used. It is also unclear why the authors did not use the LM2 cell lines employed in the humanized mouse model in the initial experiments. The authors should extend the studies to other tumors to reach general conclusions, as those stated in the title and abstract.

2. In several experiments, authors show "a representative experiment out of two independent experiments". At least three independent experiments are required to reach conclusions and the figures should represent means of independent experiments and not technical replicates. In other experiments, such as Supp Fig S8, results derive from just one experiment performed in T cells from 3 mice or even just a single mouse. Figure legends in many cases do not contain enough details to understand the experiments (ie, what is the timepoint in Fig 6a, when was the treatment initiated?)

Other points:

3. Why the authors use the expression of PIM as a marker of hypoxia in the murine 4T1 model and not the expression of HIF1 α , as done in the human model? Is there a correlation between PIM and HIF1 α expression? An IHC staining of HIF1 α should be performed. Moreover, in Figure 6g, when cells positive for HIF1 α disappear, a high percentage of PIM-positive cells remain. Also, in IHC images, PIM expression is not altered in response to ENT + anti-PD1 or PX478 + anti-PD1. Thus, it is unclear that PIM and HIF1 α are comparable to measure hypoxia. HIF1 α , and not PIM, should be used in the mouse model.

4. Supplemental Figure 2b and 2c are not mentioned in the text, and 2c referred in the text correspond with 2d in the figure.

In this same figure, authors claim that hypoxia slightly decreased the T_{em} population, but also increases the T_{eff} population, this must be indicated. Also, the sentence "Unlike T cells, NK cells coculturing with TNBC did not further induce TIM-3 and TIGIT expression (Fig S2h)" is not completely certain because TIM-3 is induced. Regarding this, authors state: "Taken together, we have demonstrate that hypoxia, together with antigen stimulation, suppressed immune effector gene expression and induced TIM-3 and TIGIT in both T and NK cells", but, in NK cells TIGIT is not induced.

5. Authors conclude that persistent antigen stimulation during co-culture with TNBC further amplified the exhaustion signals to intensify the T cell exhaustion. What exactly does the co-culture with TNBC compared to monoculture? What is the advantage of adding TNBC cells? Because the downregulation of immune effectors, the increase in TIM-3 and TIGIT in CD8⁺ cells and the percentage of PD1⁺TIM3⁺ CD8⁺ cells is similar in both conditions. This question should be clarified because authors state that antigen stimulation is necessary to perform this culture model.

5. Results of TNF promoter are not shown in Figure 4a, 4d and 4e.

6. The quality of the immunoblot in figure 4g is poor, the inhibition of HIF1 α expression is not observed in these conditions.

7. In Figure 5b and 5d, there are discrepancies in the expression of pSTAT1 and IRF1 in the condition without T cells in MDA-MB-231, what is the reason? In figure 5d, immunoblots of BT549 should be shown since the increase in pSTAT1 is clearer than in MDA-MB-231. IFN γ treated cells should be included as positive control of the signaling.

8. In Figure 6e, there are not dots in the T depletion condition at D30, why?

9. In the 4T1 model, hypoxia does not increase the expression of TIGIT and the treatment with EPX + anti-PD1 do not improve the effect in metastasis, as authors claim.

Minor points

1. Improve Figure 1f; it is not possible to see IFN γ staining (red cells) of the 3rd panel
2. Include Luminal B subtype in the Supplementary Figure 1a
3. Explain better in the figure legend the experimental design of Figure 2c
4. Figure 2i: Please confirm conditions of the experiment: resting, mono o coculture
5. Supplemental Figure 2: Letters a-h are duplicated, and are not correctly referenced in the text (ie Supp Fig 2c).
6. Supplementary Figure 2d, please include the 4 donors as in the other panels.
7. In Figure 2b, 3g, 4f, 6b, 7c, Supl.4c and Supl.6a, it is not possible differentiate points and lines in graphics due to the colors used
8. Page 7, end of the first paragraph: "T cells differentiation" should be "T cell differentiation".
9. Supplemental Figure 6b: The statistical significance comparing controls is missing.
10. Supplemental Figure 6f: "CD8+ T cells infiltration" should be "CD8+ T cell infiltration".
11. Supplemental Figure 7a: Are the NK cells from donor 1 or 2? Both donors should be shown.
12. Revise letter font for Supplementary Figure 6.
13. In the page 6 line 4 it is missing an "and" between (Teff) and CD45RO+CD62L-
14. Please indicate acronyms first time they appear (ie PBMC, page 5 line 22).
15. Page 12 line 6 the word *in vivo* is not in italics.

Reviewer #4 (Remarks to the Author): with expertise in T cell dysfunction, epigenetics

In this study, Ma et al. use an *in vitro* co-culture system in hypoxia to study HIF1a dependent regulation of T cell function. The topic is of great interest to the field, however, some methodological limitations and concerns about novelty limit enthusiasm for the manuscript as it stands.

1) The authors imply that they have created an *in vitro* model of T cell exhaustion. Unfortunately, this claim is overly strong given the data they present. For example, exhaustion is associated with specific transcriptional and epigenetic changes and it is not clear that this *in vitro* model recapitulates all or some of those changes. The expression of multiple activation/inhibitory receptors does not necessarily imply these changes are related to exhaustion. Of note, the dysfunction described by the authors seem to be much more related to hypoxia than T cell exhaustion since changes in IFN γ , TNF α , and Gzmb are much more driven by changes in oxygen levels, rather than the presence of tumor. If the authors suggest they have recreated aspects of exhaustion, they need to do much more thorough profiling of CD8 T cells in this system and compare back to published *in vivo* datasets to confirm similar changes.

2) In the entire study, the authors do not account for the fact that substantial heterogeneity exists within tumor reactive CD8+ T cells *in vivo* and potentially in their *in vitro* system. For example, does gating on multiple inhibitory receptors change the functionality of CD8+ T cells under the various genetic and pharmacological perturbations? It would be critical to know whether these interventions are altering IFN γ , for example, globally in all T cells or causing expansion/contraction of specific T cell subsets. Furthermore, do any of these perturbations dramatically alter cell viability and/or proliferation? These data are critically important to properly interpret these findings.

3) Most importantly, these results directly contradict previous studies looking at the role of HIF1a in CD8+ T cells in the tumor (Liikanen et al 2021 and Palazon et al 2017). How do the authors reconcile this? The point in the discussion section that mice and humans are different seem insufficient since the authors also rely on mouse model data in Figure 6. There is a major concern that hypoxia and/or these epigenetic drugs are largely altering the tumor cells or APCs but not directly the T cells. The *in vivo* data are interesting but are somewhat correlative and the observed effects of the

drugs could be driven by different mechanisms in vivo. Of note, the work from Greg Delgoffe's lab would broadly agree with some of these findings, but their data on HIF1a also contradicts the findings in this study to some extent. At least 2 key experiments would enhance interpretation of the data presented here: that the effect of genetic loss of HIF1a can be reversed by HIF1a overexpression (rescue experiment) and that genetic loss of HDAC and/or EZH2 specifically in CD8 T cells can recapitulate the phenotype shown here (since drugs may be non-specific or acting in other cells).

Reviewer #1

Ma S et al. reported that HIF1a interacted with HDAC1 to control HDAC1 and EZH2 binding to IFNG, TNF, and GZMB genes to repress their gene expression in T cells and NK cells, leading to immune evasion and resistant to immunotherapy. Combined treatment of HDAC1 inhibitor or HIF-1 inhibitor with anti-PD1 enhanced immunotherapy in syngeneic and humanized mouse models of TNBC. Although the story is interesting, the key conclusions are not well supported by the existing data in the manuscript. Some of data are not convincing and their scientific rigor is low. Overall, this manuscript does not meet with NC standards.

Major points:

1. Fig. 1. Hypoxia is known to be uncorrelated with tumor subtypes, grades, and stages. Luminal breast tumors have less immune cell infiltration as compared with TNBC and HER2+ breast tumors. So it is not surprising that a hypoxia gene signature did not correlate with immune cell signatures in luminal breast tumors. These gene correlation data in luminal breast tumors do not support their conclusion in the manuscript that hypoxia is associated with immune escape in breast cancer.

Response: As we have mentioned in the text, TNBC and HER2 tumors are well documented with a higher level of hypoxia^{1,2,3}, where we see a negative correlation between hypoxia signal and immune infiltration signal. Luminal tumor indeed has lower TILs, which could be caused by other reasons unrelated to whether or not they are hypoxic. The focus of this study is TNBC. Luminal BC included here only serves as a comparison to indicate that hypoxia-TILs negative correlation is only seen in aggressive TNBC and HER2 tumors. We have modified the text to reflect this to make this point clearer.

2. Fig. 1h, S1. It is unclear how the authors defined “high” and “low” in the KM analysis. Given the different numbers in both groups, the authors did not use median to separate two groups, which seems biased.

Response: We used the “auto-best select cut-off” which defines the best performing cut-off as described in the KM website. Both auto-best cut-off and median are widely used to define groups in survival analysis^{4,5,6}. In fact, using the median gives rise to similar results. We will explain this in the revised legends. Thanks for pointing out this.

3. Fig. 3e, S3d-f. Hypoxia had no effect on EZH2/SUZ12 enrichment on IFNG and TNF genes, which cannot explain increased H3K27me3 occupancy on these genes. Reduction in H3K27ac is not sufficient to cause H3k27me3 increase as H3K27me3 is dynamically controlled by methyltransferase EZH2 and demethylases (KDM6A/B). Thus, the mechanism of H3K27ac-H3K27me3 interplay under hypoxia was not clearly addressed. In addition, it is unclear how strong EZH2/SUZ12 enrichment is as control IgG is missing in these experiments.

Response: We thought we had addressed this clearly in the manuscript. Let's try again: it is true that H3k27me3 does not increase by itself but requires the existence of PRC2. We showed clearly that PRC2 “pre-exists” at these loci even at hypoxia; though hypoxia did not increase PRC2 further enrichment, it does increase HDAC1, which removes H3K27ac, as a result of this change, H3K27me3 has increased accordingly due to the pre-existing PRC2. We have modified some of the text in this section to clarify it.

The machinery of PRC2-HDACs has been well studied in regulating histone methylation and acetylation. Several reports have shown that modulation of either PRC2 or HDAC1 alone was able to affect both H3K27ac and H3K27me at the same time^{7, 8, 9}, which coincide with our molecular models.

We should clarify that the PRC2 enrichment here is relative to IgG. It is around 3-5 fold enrichment. Please note that we have included both PRC2 negative target gene RPL30 and positive target gene CCND2 to demonstrate the specificity as in revised **Figure 3D**. We have also modified figure legends to clearly explain that enrichment of proteins in all CHIP experiments are presented as relative to IgG.

4. Fig.4a. There is no HRE at the IFNG promoter (P3 and P4). It is not convincing that HIF-1a can bind there.

Response: It is widely accepted that HRE is important, but neither sufficient nor essential for HIF1 α binding^{10, 11, 12}. Several reports of genome-wide analysis of HIF binding sites revealed that a large amount of HIF1 α binding loci lack a HRE site.^{10, 11} An example is that HIF1 α forms a complex with Notch 1 and binds to promoter of HEY-2 without a HRE site¹³. This is actually similar with our model that HIF1 α acts to silence gene expression via interaction with HDAC1.

5. Fig. 4e. Given that HIF-1a did not bind to EZH2 and SUZ12, how does HIF-1a KD reduce H3K27me3 enrichment?

Response: Same as question 3, HIF1 α KD impaired HDAC1 binding, resulting in increased H3K27ac, and reduced H3K27me3.

6. Fig. 5a. Tumor cells grow slowly on the petri dishes under hypoxia. Is reduced luciferase activity under hypoxia due to slow tumor cell growth?

Response: This is a killing assay of cancer cells by hypoxia pre-treated immune cells. All the cancer cell viability has already been normalized to the control group (tumor cells without immune cells, under normoxia or hypoxia). This practice is to exclude the direct effects of hypoxia on tumor cells. To further clarify this, we have some unshown data to demonstrate that hypoxia treatment of cancer cells does not affect their growth. Protocol 1 shows hypoxia pre-treatment of immune cells dampens the killing of cancer cells; protocol 2 shows hypoxia pre-treatment of cancer cells does not affect the killing by immune cells.

7. Fig. 7. The authors implanted human LM2 cells after establishment of humanized mouse model. It is surprising that these tumor cells grew similarly to those in immunodeficient mice and were not rejected in this humanized model?

Response: Looking at the tumor volume data in **Figure 7b**, the tumors in the humanized mouse grow significantly smaller than those in NIKO mice. We have added some statistical analysis to support this observation.

8. Fig. 7e. Combined treatment failed to synergize IFN γ production, which does not support their finding in tumor growth (Fig. 7b).

Response: In fact, both ketruada and ENT alone did not significantly induce IFN γ in infiltrating T cells in 231-LM2 tumors compared to control group, but the combination did. Besides, in addition to IFN γ , TNF α and granzyme B also shows the synergistic induction by both combinations. We believe these data are sufficient to support the tumor growth inhibition.

Minor points:

1. GAPDH was used as a loading control in western blot data, which is not appropriate. GAPDH is a HIF downstream target gene.

Response: In fact, we did not find GAPDH protein level change by hypoxia in our cells. Anyway, we also included Tubulin as an additional control in revised **Figure 5b**

2. Fig. 4b. HIF-1a blot (input) is not convincing.

Response: Provided a better one in revised **Figure 4b**

3. Fig. 6h. The data need to be quantified. The hypoxia area is not correctly circled in the CD8 staining image in PX478+aPD-1 group.

Thanks for pointing this out. We now corrected the figure and quantification data can be found in revised **Figure 6i**

4. A spelling error for PD-L1 on page 11, 2nd paragraph. **Corrected**

5. The authors should add page numbers. **Added**

Reviewer #2

While the understanding of complex tumor microenvironment and response to immunotherapy is required to promote therapeutic efficacy, this manuscript determines how hypoxia plays a role in immune evasion by focusing on triple negative breast cancer (TNBC). Using both in vitro and in vivo systems, authors demonstrate that the interaction between HIF and HDAC suppresses immune effector genes through epigenetic alterations. By targeting HIF and HDAC, authors suggest combination treatments with PD1 therapy, which can be possibly translated into the clinic. Overall this manuscript clearly presents their supporting data with extensive studies. However, there are minor points that need to be addressed to better present their findings.

1. Figure 1b) It is interesting that HER2 overexpressing breast cancer acts in similar way with TNBC. It might need to be mentioned what causes the differences between TNBC, HER2 and Luminal cancer types. Also, authors need to address rationale why they focus more on TNBC but not HER2 overexpressing breast cancer in terms of hypoxia-mediated T cell exhaustion unless they provide data using HER2 positive cells.

Response: TNBC and HER2 tumors are well known to be more associated with hypoxia^{2,3} that is probably why we do not see a correlation between hypoxia signaling with TILs in luminal cancer. As for why we focus on TNBC but not HER2, a major reason is that anti PD1 therapy has been approved in TNBC but with limited response rate^{14,15}. The reason for this is still unknown. By contrast, HER2 tumors have excellent targeted treatment (Herceptin) and the clinical need is perhaps not so urgent as TNBC. We are keen to understand this and aim to be translational to address the clinical need in TNBC. We have modified the text to make this point clearer.

2. Figure 1c, 1f,6h) Throughout the paper, authors did not provide any quantitative data of tissue staining. It is not clear how representative staining images are. Quantification needs to be included by comparing intensity or areas of positive staining.

Response: Figure 1d and Figure 1e are the quantitative data for Figure 1c. For the mouse tissue in Figure 1f, we have tested several anti-mouse INF γ antibodies but can not achieve INF γ staining as good as in our human TNBC samples for further quantification. We hence decided to show one picture of IF staining as a visual demonstration and we used flow cytometry to provide the quantification as shown in Figure 1g.

For figure 6h, quantification can be found in revised Figure 6i

3. Page 6. Indicated figures are mislabelled. Authors indicate Fig S2b and 2d instead of Fig S2c in this sentence. "Hypoxia did not seem to affect the CD4+ and CD8+ T cell compositions, nor did it affect the overall T cell differentiation status, though it slightly decreased the Tem population (Fig. S2c)"

Corrected. Thanks for pointing this out

4. Figure 2d. Authors need to indicate what the gene list in type 2 means.

Response: We decided to change the terms of “type” to “gene cluster” to avoid confusion. Explanations of gene cluster 1 and 2 can be found in **Fig. 2d**.

5. Figure 4a. Authors need to include positive control data for their HIF-1 and HIF-2 ChIP assays, since it is not clear whether insignificant changes in HIF-2 ChIP are due to a technical problem of poor antibody binding or an actual HIF-2-promoter binding.

Response: Thanks for this comment. Now we have added VEGFA as a positive target of both HIF1a and HIF2a in revised **Figure 4a**

6. Figure 4g. The quality of Western blot needs to be improved since it is hard to recognize the inhibition effects from HIF1 inhibitors.

A better blot image was provided in revised **Figure 4g**

7. Figure 6a, 6d, 7b, S9a. Authors present tumor growth data inconsistently. Tumor growth curve needs to be included in each figure to compare the effect of the drug combination.

We have now provided the growth curve data in revised **Supplement Figure 9a and 9e**

8. Figure 6b, 6e, S9b. It will be more comprehensive if authors include IHC data from lung showing how metastatic lung nodules formed in different ways depending on each treatment. Bioluminescence imaging is a good tool to track tumor growth non-invasively but also, background effect may need to be considered.

Response: We understand the reviewer’s concern about the background issue in bioluminescence imaging. Now images of lung samples stained with Bouin’s solutions to demonstrate the metastasis, along with the bioluminescence imaging, are shown in revised **supplementary Figure 9d**, corresponding to **Figure 6b**. All mice were sacrificed at day 30 for lung collection and staining. For Figure 6e it was challenging to do so, as T depleted mice quickly died before the control mice developed detectable lung metastasis, so unable to collect all mice samples at the same time point.

9. Figure 7b, 7c. To emphasize the effect of immune system, authors need to include treatment of PX478 and ENT alone to compare with combination treatment, since ENT and PX478 have antitumor effects on their own.

Response: We appreciate this comment. We do have single drug data in the humanized mouse. Because we are focused on validating the role of the immune system in the combination effect (and to save mice), we did not include the single treatment group in the NIKO mice. We understand the concern but we did find in other separate experiments that ENT and PX478 alone has some effects in LM2 tumor growth in NIKO mice as expected.

In another experiment, NIKO mice bearing LM2 tumors received ENT and PX478 at the same dose as indicated in the manuscript. The below data shows the tumor volume measured at Day 20.

Although it is hard to combine the single treatment data with the combination treatment data in NIKO mice (due to sample size and tumor measurement time point), we can still try to compare the tumor growth trending in two experiments. In the single treatment experiment, average tumor volume of ENT group and PX478 group is around 75% and 66% of the control group at Day 20 of treatment, respectively. In the combination treatment experiment, average tumor volume of

ENT+Keytruda and PX478+Keytruda is around 72% and 63% of the control group at Day 21 of treatment. Unlike the humanized mice, Keytruda does not show obvious combinational effects with ENT and PX478 in NIKO mice. We hope this data can address the reviewer's concern.

10. Although authors noted in the discussion section, it is not clear how hypoxia regulates TIM-3 and TIGIT and what roles they play in the context of this study. Unless authors provide further data suggesting HIF-independent mechanisms altering these factors, they are better to be excluded in this manuscript.

Response: Yes, the main focus of this paper is the HIF1a-induced effector dysfunction of immune cells. However, there are also reports showing that hypoxia can induce T exhaustion independent of HIF1a. Our data shows that hypoxia can induce T cell effector dysfunction per se via HIF1a, but hypoxia can also induce exhaustion markers expression via HIF1a-independent manner. We feel this data can help to address some discrepancies in the literature. This is also necessary to address one of the questions from reviewer 4

11. The conclusion clearly supports that combination treatment of HDAC or HIF inhibitors with PD1 therapy inhibits primary tumor growth as well as metastasis. It will be more supportive if authors include patient data showing the correlation of hypoxia vs. metastasis or immune effectors vs. metastasis in TNBC patients.

Response: We have shown that role of hypoxia and immune signatures in distant metastasis-free survival of TNBC patients. We also managed to get a few metastasis TNBC samples to analyze hypoxia and immune effectors. In fact, **Figure 1c**, TNBC tumor #3 is a stage IV tumor that shows strong hypoxia and low TIL and IFN γ . We were also trying to obtain paired primary vs metastasis TNBC tumors but only got a few pairs so we did not pursue this, though it indeed shows that metastasis tumors shows more hypoxia and low IFN γ . We did not include the data as it has no statistical power.

Reviewer #3

Hypoxia is common in several solid tumors and contributes to resistance to several therapies, including immunotherapy. Better understanding of the mechanisms governing hypoxia in the tumor microenvironment will allow identifying new targets in order to develop more precise therapeutic approaches.

In this work, Ma et al. highlight the importance of hypoxia in conferring resistance to immunotherapy in triple-negative breast cancer (TNBC). They demonstrate that under a hypoxic tumor microenvironment, tumor cells of a triple negative breast cancer model acquire resistance to immune checkpoint blockade. This process is controlled by an epigenetic mechanism that involves HIF1 α but also by a parallel HIF1 α -independent mechanism that leads to the suppression of immune effector gene expression. Using *in vitro* assays, the authors show that hypoxia and the epigenetics events associated to HIF1 α impair the immune cells cytotoxicity, the interferon signaling, and induces resistance to immunotherapy (such as promoting a resistance to anti-PD1 blockade). *In vivo* experiments show that targeting HIF1 α and the associated epigenetic machinery overcomes resistance to PD1 blockade, in both syngeneic and humanized models.

Although most results in this manuscript are consistent, and they employ a wide range of experimental settings, some conclusions are not sustained by the experimental data, and the number of independent experiments and biological replicates need to be increased in several settings.

Major issues:

1. It is unclear to me why the authors have selected breast cancer to study the contribution of hypoxia to immunotherapy response. Despite immunotherapy has been recently approved in TNBC, it will be more relevant to perform their studies in melanoma or lung cancer, where immunotherapy has become the standard of care. Analyses and experiments in these tumor types will be more relevant to the clinical practice. The title of the manuscript does not mention "breast cancer" and in the abstract the authors extrapolate the results to other solid tumors, but only one mouse and two human breast cancer cell lines were used. It is also unclear why the authors did not use the LM2 cell lines employed in the humanize mouse model in the initial experiments. The authors should extend the studies to other tumors to reach general conclusions, as those stated in the title and abstract.

Response: We appreciate the comments on the title and abstract. Indeed, this study focuses on TNBC and we should stay with TNBC in both title and abstract. We have modified the manuscript according to this comment.

Immunotherapy in lung cancer and melanoma had been extensively studied, while there is little study on TNBC immunotherapy. Given that anti-PD immunotherapy has been approved to treat TNBC but the clinical response is limited^{14,15}, we do see its value to explore the resistance mechanism in TNBC which appears to represent more unmet clinical needs, as compared to lung and melanoma. Practically speaking, my lab is a breast cancer lab and we have a long-term interest in TNBC with aims to understand more and develop effective therapy.

About LM2 cells: We actually found MB231 and MB231-LM2 behave similarly in response to immune stimulation *in vitro*. LM2 is a convenient model for *in vivo* tumor growth and metastasis study as it carries a luciferase reporter. We have now provided the LM2 cells data in **Supplementary Figure 7b. We have also modified the text to explain better why we used LM2 cells for xenograft model.**

2. In several experiments, author show “a representative experiment out of two independent experiments”. At least three independent experiments are required to reach conclusions and the figures should represent means of independent experiments and not technical replicates. In other experiments, such as Supp Fig S8, results derive from just one experiment performed in T cells from 3 mice or even just a single mouse. Figure legends in many cases do not contain enough details to understand the experiments (ie, what is the timepoint in Fig 6a, when was the treatment initiated?)

Response: Most of the data referred is CHIP data and western blot, for CHIP, because of its big variation between experiments, we did not combine the data instead we showed it as representative of two independent experiments. It is hard to combine data from different experiments for CHIP-qPCR. Acutally it is a quite common practice to present CHIP-qPCR result as mean of technical triplicates, representative of biological duplicates.^{16, 17, 18, 19}

For all the *in vitro* experiments using samples from different human donors or mice, the experiments for each sample were performed independently, instead of all samples in one experiment. To further address reviewer’s concern, two more mouse T cells were added in quantification in revised **Supplement Figure 8**.

We have also modified the figure legends to provide more details.

Other points:

3. Why the authors use the expression of PIM as a marker of hypoxia in the murine 4T1 model and not the expression of HIF1alpha, as done in the human model? Is there a correlation between PIM and HIF1alpha expression? An IHC staining of HIF1alpha should be performed. Moreover, in Figure 6g, when cells positive for HIF1alpha disappear, a high percentage of PIM-positive cells remain. Also, in IHC images, PIM expression is not altered in response to ENT + anti-PD1 or PX478 + anti-PD1. Thus, it is unclear that PIM and HIF1alpha are comparable to measure hypoxia. HIF1alpha, and not PIM, should be used in the mouse model.

Response: This needs to understand the difference between HIF1 α and PIM. PIM or pimonidazole is a small molecular probe detecting hypoxia, not a protein. You can refer to this report for the detailed mechanism of PIM²⁰. PIM marks hypoxia independently of HIF1 α (hypoxia can generate both HIF1 α dependent and independent changes). In another word, HIF1 α is a protein induced by hypoxia, which can be knock-down or targeted by compounds. PIM, as a molecular probe, can be “activated” as long as the hypoxia (low oxygen) exists. PX478 is a HIF1 α inhibitor, so depletion of HIF1 α by PX478 will not influence PIM level. Compared to HIF1 α staining, PIM staining is a more convenient and reliable method to detect hypoxia, with both IF staining or flow cytometry. We did not use it in the patient sample because it can only be used by injection to the subjects.

We have modified the text and figure legends to better explain the function of PIM to avoid confusion.

4. Supplemental Figure 2b and 2c are not mentioned in the text, and 2c referred in the text correspond with 2d in the figure.

In this same figure, authors claim that hypoxia slightly decreased the Tem population, but also increases the T_{eff} population, this must be indicated. Also, the sentence “Unlike T cells, NK cells coculturing with TNBC did not further induce TIM-3 and TIGIT expression (Fig S2h)” is not completely certain because TIM-3 is induced. Regarding this, authors state: “ Taken together, we have

demonstrate that hypoxia, together with antigen stimulation, suppressed immune effector gene expression and induced TIM-3 and TIGIT in both T and NK cells”, but, in NK cells TIGIT is not induced.

Thanks for pointing out this. We have now revised the text to be more accurate.

5. Authors conclude that persistent antigen stimulation during coculture with TNBC further amplified the exhaustion signals to intensify the T cell exhaustion. What exactly does the coculture with TNBC compared to monoculture? What is the advantage of adding TNBC cells? Because the downregulation of immune effectors, the increase in TIM-3 and TIGIT in CD8+ cells and the percentage of PD1+TIM3+ CD8+ cells is similar in both conditions. This question should be clarified because authors state that antigen stimulation is necessary to perform this culture model.

Response: Adding TNBC means further countering the antigen, which indicates a continued antigen stimulation in addition to DC stimulation, which mimics T cell exhaustion *in vivo*. As shown in Fig. 2g, coculturing with TNBC further differentiates the T cells from naïve T cells to Tem and Teff. Compared to monoculture, the coculture model is more related to tumor-infiltrating T cells. Technically speaking, although hypoxia alone (without the TNBC coculture) is sufficient to bring down the effector cytokines, hypoxia plus coculture with TNBC induces obviously stronger TIM3 and TIGIT3 expression compared to no TNBC condition (**Fig. 2f**).

5. Results of TNF promoter are not shown in Figure 4a, 4d and 4e.

Now provided in revised **Supplementary Figure 5 a, d and e**

6. The quality of the immunoblot in figure 4g is poor, the inhibition of HIF1alpha expression is not observed in these conditions.

Please see revised **Figure 4g**

7. In Figure 5b and 5d, there are discrepancies in the expression of pSTAT1 and IRF1 in the condition without T cells in MDA-MB-231, what is the reason? In figure 5d, immunoblots of BT549 should be shown since the increase in pSTAT1 is clearer than in MDA-MB-231. IFNgamma treated cells should be included as positive control of the signaling.

For Figure 5b, the discrepancies could be caused by the unequal loading. Please see better western blot in the revised **Figure 5b**. For Figure 5d, we have carefully examined the blots but did not see an obvious difference of pSTAT1 and IRF1 expression patterns in MDA-MB-231 without T cells. However, we did notice that the labels were too crowded in Figure 5d which could easily cause misidentification of the blot panels. We now optimized **Figure 5d** to make the labels more clear.

8. In Figure 6e, there are not dots in the T depletion condition at D30, why?

Response: Mice quickly died before day 30 in T depletion mice, due to faster tumor growth and metastasis

9. In the 4T1 model, hypoxia does not increase the expression of TIGIT and the treatment with EPX + anti-PD1 do not improve the effect in metastasis, as authors claim.

Response: I believe this is due to the sample size. We now added two more mice data so the n=5. The new data shows the significant induction of TIGIT by hypoxia (**Supplemental Figure 8c**). EPZ does not work well as ENT and PX478 in mouse model, which could be related to poor PD/PK issue. We have modified the claim accordingly.

Minor points

1. Improve Figure 1f; it is not possible to see IFN γ staining (red cells) of the 3rd panel

Improved

2. Include Luminal B subtype in the Supplementary Figure 1a

Revised **Supplementary Figure 1** to include luminal B

3. Explain better in the figure legend the experimental design of Figure 2c

4. Figure 2i: Please confirm conditions of the experiment: resting, mono or coculture

5. Supplementary Figure 2: Letters a-h are duplicated, and are not correctly referenced in the text (ie Supp Fig 2c).

6. Supplementary Figure 2d, please include the 4 donors as in the other panels.

7. In Figure 2b, 3g, 4f, 6b, 7c, Supl.4c and Supl.6a, it is not possible differentiate points and lines in graphics due to the colors used

8. Page 7, end of the first paragraph: "T cells differentiation" should be "T cell differentiation".

9. Supplementary Figure 6b: The statistical significance comparing controls is missing.

10. Supplementary Figure 6f: "CD8+ T cells infiltration" should be "CD8+ T cell infiltration".

11. Supplementary Figure 7a: Are the NK cells from donor 1 or 2? Both donors should be shown.

12. Revise letter font for Supplementary Figure 6.

13. In the page 6 line 4 it is missing an "and" between (Teff) and CD45RO+CD62L-

14. Please indicate acronyms first time they appear (ie PBMC, page 5 line 22).

15. Page 12 line 6 the word *in vivo* is not in italics.

Response: All the above issues from 3 to 15 have been corrected accordingly.

Reviewer #4

In this study, Ma et al. use an in vitro coculture system in hypoxia to study HIF1a dependent regulation of T cell function. The topic is of great interest to the field, however, some methodological limitations and concerns about novelty limit enthusiasm for the manuscript as it stands.

1) The authors imply that they have created an in vitro model of T cell exhaustion. Unfortunately, this claim is overly strong given the data they present. For example, exhaustion is associated with specific transcriptional and epigenetic changes and it is not clear that this in vitro model recapitulates all or some of those changes. The expression of multiple activation/inhibitory receptors does not necessarily imply these changes are related to exhaustion. Of note, the dysfunction described by the authors seem to be much more related to hypoxia than T cell exhaustion since changes in IFN γ , TNF α , and Gzmb are much more driven by changes in oxygen levels, rather than the presence of tumor. If the authors suggest they have recreated aspects of exhaustion, they need to do much more thorough profiling of CD8 T cells in this system and compare back to published in vivo datasets to confirm similar changes.

Response: We thank the review for this comment. Yes, we totally agree we should be more careful and more accurate in describing the process. In addition to downregulation of immune effectors by hypoxia alone, we did see upregulation of exhaustion markers TIM-3 and TIGIT, as well as PD-1⁺TIM-

3⁺ double positive population, induced by hypoxia in the presence of tumor cells. We agree that without carefully profiling the molecule change related to T exhaustion, it is only safe to define the process as “exhaustion-like cells” or “T dysfunction towards exhaustion”. We have thus modified the text to reflect this comment.

2) In the entire study, the authors do not account for the fact that substantial heterogeneity exists within tumor reactive CD8⁺ T cells in vivo and potentially in their in vitro system. For example, does gating on multiple inhibitory receptors change the functionality of CD8⁺ T cells under the various genetic and pharmacological perturbations? It would be critical to know whether these interventions are altering IFN γ , for example, globally in all T cells or causing expansion/contraction of specific T cell subsets. Furthermore, do any of these perturbations dramatically alter cell viability and/or proliferation? These data are critically important to properly interpret these findings.

Response: We have shown in **Supplement Figure 2e** that hypoxia suppressed IFN γ expression in different subtypes of CD8 T cells, including memory T cells and effector T cells.

Considering this suggestion, we have further analyzed IFN γ in different gateings of PD1 and Tim3. The new data shows that IFN γ was downregulated under hypoxia and rescued by the drug treatments in different CD8 subtypes (PD1⁺Tim3⁻, or PD1⁺Tim3⁺) as shown in **Supplemental Figure 6f**. We also have added new data in **Supplementary Figure 6d** to show that these perturbations do not influence the T cell viability.

3) Most importantly, these results directly contradict previous studies looking at the role of HIF1a in CD8⁺ T cells in the tumor (Liikanen et al 2021 and Palazon et al 2017). How do the authors reconcile this? The point in the discussion section that mice and humans are different seem insufficient since the authors also rely on mouse model data in Figure 6. There is a major concern that hypoxia and/or these epigenetic drugs are largely altering the tumor cells or APCs but not directly the T cells. The in vivo data are interesting but are somewhat correlative and the observed effects of the drugs could be driven by different mechanisms in vivo. Of note, the work from Greg Delgoffe's lab would broadly agree with some of these findings, but their data on HIF1a also contradicts the findings in this study to some extent. At least 2 key experiments would enhance interpretation of the data presented here: that the effect of genetic loss of HIF1a can be reversed by HIF1a overexpression (rescue experiment) and that genetic loss of HDAC and/or EZH2 specifically in CD8 T cells can recapitulate the phenotype shown here (since drugs may be non-specific or acting in other cells).

Response: Although there are previous reports showing a role of hypoxia in enhancing CD8 T cell function as the reviewer pointed out and also mentioned in the manuscript^{21, 22, 23}, there are also many other reports supporting a role of hypoxia or HIF1a in immune escape in cancer^{24, 25, 26, 27}. Both literatures have been cited in the manuscript. Although the reason for this discrepancy remains unknown, our data is in agreement with many clinical observations that hypoxia is correlated with immune escape and resistance²⁸. Most interestingly, a very recent study has shown that ectopic HIF1 α overexpression decreased IFN γ expression in T cells upon stimulation and impaired antitumor immunity²⁹, though there is no mechanistic investigation. This study is entirely in line with our findings and supports the role of HIF1 α in promoting immune escape.

In Dr. Greg Delgoffe's recent paper³⁰, they focused on hypoxia and tumor antigen on T cell exhaustion and demonstrate the HIF1 α is not required for this process. However, the authors addressed this question very carefully. They claimed that they could not totally exclude the role of HIF1 α in the hypoxia-driven T cell exhaustion (4th paragraph in the Discussion part). Looking at their

data carefully, HIF1 α KD does not affect exhaustion markers expression (Fig S4a), but seems to increase the IFN γ level slightly under hypoxia (Fig S4b), although they did not further elaborate this with statistical analysis. This is actually consistent with our observation and hypothesis that hypoxia requires HIF1 α to induce downregulation of immune effectors, but hypoxia induced T exhaustion (such as TIM3, TIGIT induction) could be HIF1 α -independent.

As for the concern that hypoxia and epigenetic drugs may have effects on tumor cells or APC but not directly on T cells, our large amount *in vitro* data has already addressed they work on T cells directly. For example, we only pre-treated the immune cells with drugs and washed the drugs away before coculture to directly exclude the influence on tumor cells. New data in **Figure 7h** also shows that hypoxia and these drugs did not affect the tumor cells viability in conditions used in the immune cell killing assay. Of course, *in vivo*, it is challenging to exclude their effect on cells other than T cells, and we do not exclude other effects of these drugs *in vivo* as mentioned in the discussion.

To substantiate the conclusion, we performed rescue experiments in T cells as requested by the reviewer. The new data has demonstrated that HIF1 α overexpression effectively abolished the immune effector upregulation upon HIF1 α KD (**Supplementary Figure 6a, 6b**). In addition, data from knockdown of HDAC1 and EZH2 in T cells also been provided in **Figure 3 and 4**.

Reference

1. Comprehensive molecular portraits of human breast tumours. *Nature* **490**, 61-70 (2012).
2. Tutzauer J, *et al.* Breast cancer hypoxia in relation to prognosis and benefit from radiotherapy after breast-conserving surgery in a large, randomised trial with long-term follow-up. *British Journal of Cancer*, (2022).
3. Ye IC, Fertig EJ, DiGiacomo JW, Considine M, Godet I, Gilkes DM. Molecular Portrait of Hypoxia in Breast Cancer: A Prognostic Signature and Novel HIF-Regulated Genes. *Molecular Cancer Research* **16**, 1889-1901 (2018).
4. Huang H, *et al.* HNRNPK inhibits gastric cancer cell proliferation through p53/p21/CCND1 pathway. *Oncotarget* **8**, (2017).
5. Yu P, *et al.* PKM2–c-Myc–Survivin Cascade Regulates the Cell Proliferation, Migration, and Tamoxifen Resistance in Breast Cancer. *Frontiers in Pharmacology* **11**, 550469 (2020).
6. Meng Y-M, *et al.* Hexokinase 2-driven glycolysis in pericytes activates their contractility leading to tumor blood vessel abnormalities. *Nature Communications* **12**, 6011 (2021).
7. Li F, *et al.* Histone Deacetylase 1 (HDAC1) Negatively Regulates Thermogenic Program in Brown Adipocytes via Coordinated Regulation of Histone H3 Lysine 27 (H3K27) Deacetylation and Methylation. *The Journal of biological chemistry* **291**, 4523-4536 (2016).
8. Pasini D, *et al.* Characterization of an antagonistic switch between histone H3 lysine 27 methylation and acetylation in the transcriptional regulation of Polycomb group target genes. *Nucleic Acids Res* **38**, 4958-4969 (2010).
9. Nguyen TTT, *et al.* HDAC inhibitors elicit metabolic reprogramming by targeting super-enhancers in glioblastoma models. *The Journal of clinical investigation* **130**, 3699-3716 (2020).
10. Mole DR, *et al.* Genome-wide association of hypoxia-inducible factor (HIF)-1alpha and HIF-2alpha DNA binding with expression profiling of hypoxia-inducible transcripts. *The Journal of biological chemistry* **284**, 16767-16775 (2009).
11. Xia X, *et al.* Integrative analysis of HIF binding and transactivation reveals its role in maintaining histone methylation homeostasis. *Proceedings of the National Academy of Sciences* **106**, 4260-4265 (2009).
12. Dengler VL, Galbraith M, Espinosa JM. Transcriptional regulation by hypoxia inducible factors. *Crit Rev Biochem Mol Biol* **49**, 1-15 (2014).

13. Gustafsson MV, *et al.* Hypoxia requires notch signaling to maintain the undifferentiated cell state. *Developmental cell* **9**, 617-628 (2005).
14. Schmid P, *et al.* Pembrolizumab for Early Triple-Negative Breast Cancer. *New England Journal of Medicine* **382**, 810-821 (2020).
15. Nanda R, *et al.* Pembrolizumab in Patients With Advanced Triple-Negative Breast Cancer: Phase Ib KEYNOTE-012 Study. *J Clin Oncol* **34**, 2460-2467 (2016).
16. Semer M, *et al.* DNA repair complex licenses acetylation of H2A.Z.1 by KAT2A during transcription. *Nature Chemical Biology* **15**, 992-1000 (2019).
17. Ji Y, *et al.* miR-155 harnesses Phf19 to potentiate cancer immunotherapy through epigenetic reprogramming of CD8+ T cell fate. *Nature Communications* **10**, 2157 (2019).
18. Serio J, *et al.* The PAF complex regulation of Prmt5 facilitates the progression and maintenance of MLL fusion leukemia. *Oncogene* **37**, 450-460 (2018).
19. Sachs P, *et al.* SMARCAD1 ATPase activity is required to silence endogenous retroviruses in embryonic stem cells. *Nature Communications* **10**, 1335 (2019).
20. Varia MA, *et al.* Pimonidazole: a novel hypoxia marker for complementary study of tumor hypoxia and cell proliferation in cervical carcinoma. *Gynecologic oncology* **71**, 270-277 (1998).
21. Palazon A, *et al.* An HIF-1 α /VEGF-A Axis in Cytotoxic T Cells Regulates Tumor Progression. *Cancer cell* **32**, 669-683.e665 (2017).
22. Roman J, *et al.* T-Cell Activation under Hypoxic Conditions Enhances IFN- γ Secretion. *American Journal of Respiratory Cell and Molecular Biology* **42**, 123-128 (2010).
23. Gropper Y, Feferman T, Shalit T, Salame T-M, Porat Z, Shakhar G. Culturing CTLs under Hypoxic Conditions Enhances Their Cytotoxicity and Improves Their Anti-tumor Function. *Cell Reports* **20**, 2547-2555 (2017).
24. Lukashev D, *et al.* Cutting edge: hypoxia-inducible factor 1 α and its activation-inducible short isoform I.1 negatively regulate functions of CD4+ and CD8+ T lymphocytes. *Journal of immunology (Baltimore, Md : 1950)* **177**, 4962-4965 (2006).
25. Scharping NE, Menk AV, Whetstone RD, Zeng X, Delgoffe GM. Efficacy of PD-1 Blockade Is Potentiated by Metformin-Induced Reduction of Tumor Hypoxia. *Cancer immunology research* **5**, 9-16 (2017).

26. Ni J, *et al.* Single-Cell RNA Sequencing of Tumor-Infiltrating NK Cells Reveals that Inhibition of Transcription Factor HIF-1 α Unleashes NK Cell Activity. *Immunity* **52**, 1075-1087.e1078 (2020).
27. Thiel M, *et al.* Targeted deletion of HIF-1alpha gene in T cells prevents their inhibition in hypoxic inflamed tissues and improves septic mice survival. *PloS one* **2**, e853 (2007).
28. Hugo W, *et al.* Genomic and Transcriptomic Features of Response to Anti-PD-1 Therapy in Metastatic Melanoma. *Cell* **165**, 35-44 (2016).
29. Veliça P, *et al.* Modified Hypoxia-Inducible Factor Expression in CD8+ T Cells Increases Antitumor Efficacy. *Cancer immunology research* **9**, 401-414 (2021).
30. Scharping NE, *et al.* Mitochondrial stress induced by continuous stimulation under hypoxia rapidly drives T cell exhaustion. *Nature Immunology* **22**, 205-215 (2021).

REVIEWER COMMENTS

Reviewer #1 (Remarks to the Author):

The authors addressed many of previous concerns. However, some of responses lack experimental evidence as discussed below.

Fig.3e,S3d-f,4e. The authors' argument is purely based on their speculation. Pre-existence of the PRC2 complex on the IFNG promoter does not mean that the complex is functional. H3K27 can be modified (acetylated, methylated, crotonylated, formylated, etc) or unmodified. H3K27ac loss is not equal to H3K27me3 gain, and vice versa. The switch of H3K27 modifications is not as simple as the authors speculated. If authors' speculation is true, why does the PRC2 complex not increase H3K27me3 levels on P1, where H3K27ac levels are relatively low? Moreover, H3K27ac levels are decreased under 1% O₂ on P2, but HDAC1/2 enrichment on P2 is not altered by hypoxia. Neither of these results nicely support the conclusion. If authors want to claim this strong conclusion, they should additionally measure the activities of the PRC2 complex and HDACs under 20% vs. 1% O₂.

Fig. 4a. Numerous in vitro and in vivo studies including studies the authors cited all revealed that the direct DNA binding of HIF-1 and/or HIF-2 requires an HRE. Indeed, some of HIF-1/2alpha ChIP peaks lack the HRE, which is due to their indirect binding mediated by another protein or complex. If HIF-1alpha binding to IFNG and TNF genes is through the interaction of HDAC1, why does HIF-1alpha KD reduce HDAC1 binding? These data are not reconciled. Does HDAC1 KD also inhibit HIF-1alpha binding to the IFNG promoter?

Reviewer #2 (Remarks to the Author):

Authors responded well to reviewer's concerns.

Reviewer #3 (Remarks to the Author):

Thanks for providing "track changes" in the manuscript.

For the experiments "representative of biological replicates", authors could include the replicates as additional materials for revision.

I do understand the difference between PIM and HIF1a, and the fact that PIM cannot be used in humans. What I requested was to analyze HIF1a in the mouse models; this will help comparing human and mouse results.

Reviewer #4 (Remarks to the Author):

I commend the authors for completing additional work to address my original concerns as well as the clarifications they have made to the text/figures. I now support publication of this manuscript.

Reviewer #1 (Remarks to the Author):

The authors addressed many of previous concerns. However, some of responses lack experimental evidence as discussed below.

Fig.3e,S3d-f,4e. The authors' argument is purely based on their speculation.

Response: Please note that our argument is not based only on the ChIP data alone, which could be superficial and correlative, but also backed up by a series of interference experiments, including either knockdown of HIF1a, HDAC1 and EZH2 KD, or treated with the respective small molecule inhibitors. We believe we have provided reliable and comprehensive data to support our conclusions that hypoxia-induced HDAC1 and EZH2 engagements and the corresponding histone modifications contribute to chromatin reprogramming at immune effector genes upon hypoxia.

Pre-existence of the PRC2 complex on the IFNG promoter does not mean that the complex is functional.

Response: Again, we did not just provide the ChIP data to show PRC2 pre-existence but also performed EZH2 inhibitor treatment to demonstrate the PRC2 presence is "functional" as inhibition of its activity suppressed H3K27me3 enrichment at the *IFNG* promoter (Fig. 3f).

H3K27 can be modified (acetylated, methylated, crotonylated, formylated, etc) or unmodified. H3K27ac loss is not equal to H3K27me3 gain, and vice versa. The switch of H3K27 modifications is not as simple as the authors speculated.

Response: We totally agree that H3K27ac loss is not equal to H3K27me3. But we believe that it is undeniable that H3K27ac loss is able to lead H3K27me3 gain. We also believe our model of increased HDAC1 (H3K27ac loss), working together with functional pre-existing PRC2, is able to well explain the increased H3K27me3 on IFNG. We never speculated that H3K27 modification is simple. We understand the complexity of histone modification and we hope the reviewer can understand that we are unable to include all these modifications(crotonylated or formylated) in one study.

If authors' speculation is true, why does the PRC2 complex not increase H3K27me3 levels on P1, where H3K27ac levels are relatively low? Moreover, H3K27ac levels are decreased under 1% O2 on P2, but HDAC1/2 enrichment on P2 is not altered by hypoxia. Neither of these results nicely support the conclusion. If authors want to claim this strong conclusion, they should additionally measure the activities of the PRC2 complex and HDACs under 20% vs. 1% O2.

Response: Please note that P1 has no HIF1a and HDAC1 enrichment upon hypoxia so it is not expected to see an increase in H3K27me3 as there is no decrease in H3K27ac.

So why does P2 has no HDAC1 and HIF1a enrichment but it shows decreased H3K27ac ? We hope to clarify that P2 is only 500bp away from P3 and P4 where HDAC1 and HIF1a are strongly enriched. We know that binding patterns for histone-modifying factors and corresponding histone markers are not always aligned precisely. The former often show sharp and focused peaks, while the latter are well known to exhibit "blanket" binding patterns, which are more spread out into adjacent regions due to chromatin opening up. I put up an example here to illustrate this point.

Different binding patterns between HDAC1, which is sharp and focused and H3K27ac, which is more spread out like blanket banding

In our case, P2 is only 0.5 kb away from P3 and P4. So it is not surprised that we see HDAC1 and HIF1a binding at P3 and P4 but not P2, while H3K27ac decrease can be seen in P2, P3 and P4.

As the arrows show, the region shows the H3K7ac marker but not necessarily the HDAC1. I hope this helps to clarify the concern. The HDAC1 and PRC12 activity is reflected by the corresponding histone markers. On top of that, HDAC1 and PRC2 inhibition alter these histone markers, further consolidating the attributes of PRC2 and HDAC1 “activities” in this regulation.

Fig. 4a. Numerous in vitro and in vivo studies including studies the authors cited all revealed that the direct DNA binding of HIF-1 and/or HIF-2 requires an HRE. Indeed, some of HIF-1/2alpha ChIP peaks lack the HRE, which is due to their indirect binding mediated by another protein or complex. If HIF-1alpha binding to IFNG and TNF genes is through the interaction of HDAC1, why does HIF-1alpha KD reduce HDAC1 binding? These data are not reconciled. Does HDAC1 KD also inhibit HIF-1alpha binding to the IFNG promoter?

Response: I think there is some misunderstanding of the language here. By checking back on the language we used in the manuscript, we actually stated that hypoxia induced the interaction of HIF1a and HDAC1, leading to their increased co-occupancy at IFNG gene promoters. HIF1a knockdown reducing the HDAC1 binding to the gene promoter suggests that HIF1a induction is responsible for increased HDAC1 occupancy. We agree with this reviewer that we do not exclude the possibility that HIF1a may bind to the gene promoter through other proteins, which may recruit HDAC1 indirectly to the gene promoter. HIF1a knockdown is expected to reduce HDAC1 binding to the gene promoter in either situation. To make it more clear, we have modified some of the languages as yellow highlighted in the revised manuscript.

REVIEWERS' COMMENTS

Reviewer #4 (Remarks to the Author):

I support publishing this manuscript.